# INFINITY SEARCH: APPROXIMATE VECTOR SEARCH WITH PROJECTIONS ON Q-METRIC SPACES

## ABSTRACT

An ultrametric space or infinity-metric space is defined by a dissimilarity function that satisfies a strong triangle inequality in which every side of a triangle is not larger than the larger of the other two. We show that search in ultrametric spaces has worst-case logarithmic complexity. Since datasets of interest are not ultrametric in general, we employ a projection operator that transforms an arbitrary dissimilarity function into an ultrametric space while preserving nearest neighbors. We further learn an approximation of this projection operator to efficiently compute ultrametric distances between query points and points in the dataset. We proceed to solve a more general problem in which we consider projections in $q$-metric spaces – in which triangle sides raised to the power of $q$ are smaller than the sum of the $q$-powers of the other two. Notice that the use of learned approximations of projected $q$-metric distances renders the search pipeline approximate. We show in experiments that increasing values of $q$ result in faster search but lower recall. Overall, search in q-metric spaces is competitive with existing search methods.

## 1 INTRODUCTION

Given a dataset of vector embeddings, a dissimilarity function and a query, nearest neighbor search refers to the problem of finding the point in the dataset that is most similar to the query, or, in its stead, a vector that is among the most similar (Indyk and Motwani, 1998). It is well known that exact search requires comparison with all the points in the dataset in the worst case (Hanov, 2011) but it is also well known that this search complexity can be reduced with proper organization of the data (Malkov and Yashunin, 2020).

In particular, if we consider search problems in metric spaces, namely, problems in which the dissimilarity function satisfies the triangle inequality, data can be organized in a metric tree that can be searched with smaller average complexity (Uhlmann, 1991). In this paper we observe that metric spaces are a particular case of a more generic family of $q$-metric spaces. These spaces satisfy more restrictive triangle inequalities in which the $q$-th power of each side of a triangle is smaller than the sum of the $q$th powers of the other two sides. In the limit of growing $q$ we obtain $\infty$-metric spaces in which each side of a triangle is smaller than the maximum of the other two. The first contribution of this paper is to show that:

**(C1)** The number of comparisons needed to find the nearest neighbor of a query in an $\infty$-metric space is, at most, the ceiling of the base-2 logarithm of the size of the dataset (Section 2).

This fact is not difficult to establish. It holds because in an $\infty$-metric space each comparison in a metric tree discards half of the dataset (Proof of Theorem 1). It is nevertheless remarkable because it is an exact bound (not an order) on worst case search performance (not average). We point out that $\infty$-metric spaces are often called ultrametrics and the $\infty$-triangle inequality is often called the strong triangle inequality (Dovgoshey, 2025). Ultrametrics are equivalent to dendrograms used in hierarchical clustering (Draganov et al., 2025), a fact that may help to understand why search in ultrametric spaces has logarithmic complexity.

Because of (C1) we would like to solve search problems in $\infty$-metric spaces, thus the title of this paper. However, the data is what the data is and most problems in vector search involve dissimilarity functions that are not even metric (Zezula et al., 2006). Because of this, we seek to develop a

projection operator that we can use to map a given dataset into a general $q$-metric space, including an $\infty$-metric space. Our second contribution comes from interpreting a dataset as a graph, allowing us to leverage results from hierarchical clustering and metric representations of network data (Carlsson et al., 2014; Smith et al., 2016; Carlsson et al., 2017; Segarra et al., 2020).

In this context, it follows that the $q$-norm of a path is the $q$-root of the sum of the distances in each hop elevated to the power of $q$, which in the limit of an $\infty$-metric space reduces the maximum of the dissimilarities in the path. We emphasize for clarity that the $q$-norm of a path has no relationship with the $q$-norm of the vector embeddings. As all norms do, the $q$-norm of the vector embeddings satisfies the standard triangle inequality (and not a stronger version of it).

The canonical $q$-metric projection generates a $q$-metric space but there may be –indeed, we can construct – other possible projection operators. To argue in favor of canonical projections we draw two axioms from the existing literature (Carlsson et al., 2017). The Axiom of Projection states that the projection of a $q$-metric space should be the same $q$-metric space. The Axiom of Transformation states that the projection of a $q$-metric space must respect the partial ordering of original spaces. I.e., if the dissimilarities of a dataset dominate *all* the dissimilarities of another, the $q$-metric distances of the projected datasets must satisfy the same relationship. Our next contribution is a theoretical result that follows from this axiomatic requirement:

**(C2)** The canonical projection (which satisfies the Axioms of Projection and Transformation) preserves the nearest neighbor of any given query (Section 3.1).

As per Segarra et al. (2015), the canonical projection is the only reasonable method that we can use to generate a $q$-metric space. This is to the extent that the Axioms of Projection and Transformation are reasonable, but it is difficult to argue that they are not. Regardless, (C2) states that the canonical projection preserves the nearest neighbor information which is the focus of our search problem.

The combination of contributions (C1)-(C2) dictates that we can project datasets into $\infty$-metric spaces with a canonical projection to search with logarithmic complexity. The hitch is that $q$-canonical projections require computation of $q$-shortest paths. This is not a problem for the dataset, but it is problem for the query. To compute $q$-distances between the query and the dataset we need to compare the query with all points in the dataset, defeating the purpose of using $\infty$-metric distances to reduce search complexity. We work around this problem with our third contribution:

**(C3)** We learn a function to map vector embeddings into separate embeddings such that their Euclidean distances estimate the distances of the $q$-metric projections of the original embedding. We train this function on the dataset and generalize it to queries (Section 4)

Combining (C1)-(C3) we propose to use canonical $q$-metric projections of a dataset to generate spaces where we expect nearest neighbor search to be more efficient. We incur this cost once and reutilize it for all subsequent searches. We then use the learned embedding in (C3) to estimate $q$-metric distances between queries and points in the dataset. The use of approximate computation of $q$-metric projections renders the resulting search algorithm approximate. Numerical experiments indicate that this approach is competitive with state-of-the-art approximate search algorithms in terms of search complexity and recall (Section 5).

We refer the reader to Appendix A for a discussion of related work.

## 2  NEAREST NEIGHBOR SEARCH AND METRIC STRUCTURE

We are given a set $X$ containing $m$ vectors $x \in \mathbb{R}^n$ along with a nonnegative dissimilarity function $d : \mathbb{R}^n \times \mathbb{R}^n \to \mathbb{R}_+$ so that smaller values of $d(x, y)$ represent more similarity between points $x$ and $y$. The set $X$ along with the set $D$ containing all dissimilarities $d(x, y)$ for all points $x, y \in X$ defines a fully connected weighted graph $G = (X, D)$. We assume here that $d(x, y) = d(y, x)$ for all $x, y \in \mathbb{R}^n$, which, in particular, implies that the graph $G$ is symmetric. Examples of dissimilarity functions used in vector search are the Euclidean distance, Manhattan (1-norm) distance, cosine dissimilarity, and dissimilarities based on the Jaccard index (Zezula et al., 2006); see E.4.

We are further given a vector $x_o$ which is not necessarily an element of $X$ but for which it is possible to evaluate the dissimilarity function $d(x_o, x)$ for all $x \in X$. The nearest neighbor of $x_o$ in the set $X$

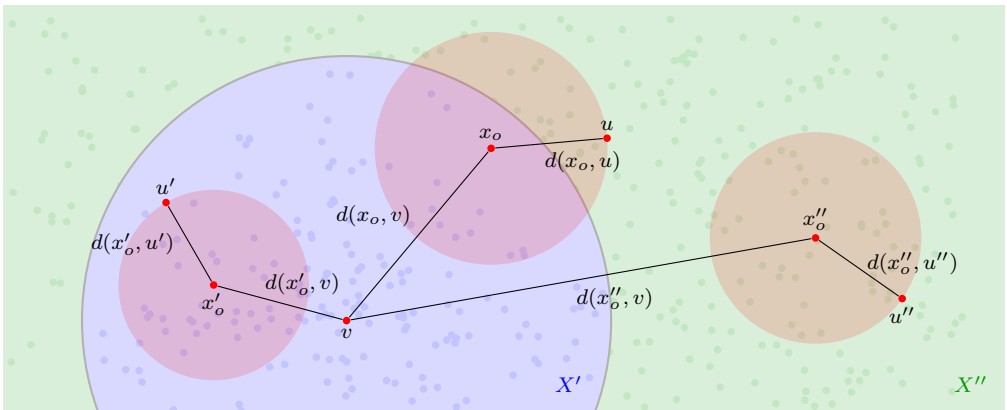

Figure 1: Search in metric spaces requires fewer comparisons than search in arbitrary spaces because for *some* queries – such as $x'_o$ and $x''_o$ – the triangle inequality allows us to restrict comparisons to subsets of the dataset $X$. Queries, such as $x_o$, for which triangle inequality bounds are inconclusive, also exist (5). This latter eventuality is impossible when the strong triangle inequality (3) holds, resulting in worst-case logarithmic complexity for search in ultrametric spaces (Theorem 1).

is defined as the element that is most similar to $x_o$,

$$\hat{x}_o \equiv \operatorname*{argmin}_{x \in X} d(x, x_o). \tag{1}$$

The problem of finding *a* vector $\hat{x}_o$ is termed nearest neighbor search, the vector $x_o$ is called a query and the vector $\hat{x}_o$ is the query's answer (Wang et al., 2014). It may be that there are several nearest neighbors. In such case we overload $\hat{x}_o$ to denote an arbitrary nearest neighbor and also the set of all nearest neighbors. We make the distinction clear when needed.

We evaluate the performance of a search algorithm by the number $c(x_o)$ of comparisons against elements of $X$ that are needed to find $\hat{x}_o$. Without further assumptions on the dissimilarity function $d$, we have $c(x_o) = m$ because we need to compare the query $x_o$ to all points in $X$. This number of comparisons can be reduced if we assume that $d$ has some metric structure (Aksoy and Oikhberg, 2010). For instance, it may be that $d$ is a proper metric or pseudometric that satisfies the triangle inequality so that for any three points $x, y, z \in \mathbb{R}^n$ we have that

$$d(x, y) \leq d(x, z) + d(y, z). \tag{2}$$

Alternatively, we may consider ultrametric dissimilarity functions $d$ (Simovici et al., 2004). In this case, triplets of points $x, y, z \in \mathbb{R}^n$ satisfy the strong triangle inequality,

$$d(x, y) \leq \max \Big[ d(x, z), d(y, z) \Big]. \tag{3}$$

In this paper, we also have an interest in $q$-metric spaces (Greenhoe, 2016). These spaces arise when the dissimilarity function is such that any three points $x, y, z \in \mathbb{R}^n$ satisfy the $q$-triangle inequality[1],

$$d^q(x, y) \leq d^q(x, z) + d^q(y, z), \tag{4}$$

for some given $q \geq 1$. For $q = 1$, (4) reduces to (2) and as $q \to \infty$, (4) approaches (3). Thus, we can think of $q$-metric spaces as interpolations between regular metric spaces that sastify the standard triangle inequality and ultrametric spaces that satisfy the strong triangle inequality. Henceforth, we may refer to (2) as the 1-triangle inequality and to (3) as the $\infty$-triangle inequality.

Figure 1 illustrates the use of metric structures in search. In this Figure, we have separated the dataset $X$ into sets $X'$ and $X''$ that contain points whose distance to some *vantage point* $v$ is smaller or larger than a threshold $\mu$ – see Appendix B for details. We further consider queries $x_o$, $x'_o$, and $x''_o$ for which we have evaluated dissimilarities $d(x_o, u)$, $d(x'_o, u')$, and $d(x''_o, u'')$ to the corresponding points $u$, $u'$, and $u''$. If it turns out that the distance $d(x'_o, v)$ between the query $x'_o$ and the vantage point $v$ is such that $d(x'_o, v) + d(x'_o, u') \leq \mu$, the diagram makes it apparent that the nearest neighbor

---

[1]Notice that the $q$-norm $||x||_q$ for $x \in \mathbb{R}^n$ satisfies the regular triangle inequality, *not* the $q$-triangle inequality.

$\hat{x}'_o$ of the query $x'_o$ *cannot* be an element of $X''$. Thus, we can restrict the nearest neighbor search to points $x' \in X'$ and save the computational cost of evaluating distances to points in $X''$. Likewise, if it turns out that for point $x''_o$ we have $d(x''_o, v) - d(x''_o, u'') \geq \mu$ we can restrict the nearest neighbor search to points $x'' \in X''$ and save the cost of evaluating distances to points in $X'$ – we give a formal proof of these statements in Proposition 2.

While this argument seems to indicate logarithmic complexity of nearest neighbor search in metric spaces, this is not quite so. The reason is that we may have queries such as $x_o$ in Figure 1 for which neither condition is true. I.e., there may be points $x_o$ such that

$$\mu < d(x_o, v) + d(x_o, u) \quad \text{and} \quad d(x_o, v) \leq \mu + d(x_o, u). \tag{5}$$

If $d(x_o, v)$ is such that (5) holds, the nearest neighbor $\hat{x}_o$ may be in $X'$ or $X''$ and we therefore do not have a reduction in the number of comparisons needed to find $\hat{x}_o$.

The argument we build in Figure 1 can be generalized to $q$-triangle inequality comparisons in $q$-metric spaces. I.e., we consider inequalities analogous to those in (5) in which distances and thresholds are $q$-powers of the corresponding quantities,

$$\mu^q < d^q(x_o, v) + d^q(x_o, u). \quad \text{(6A)} \qquad d^q(x_o, v) \leq \mu^q + d^q(x_o, u). \quad \text{(6B)}$$

If we are searching in a $q$-metric space, the exact same arguments of metric spaces are true: At least one of (6A) or (6B) must hold. If (6A) does *not* hold, the nearest neighbor must be in $X'$. If (6B) does *not* hold, the nearest neighbor must be in $X''$. If both hold, the nearest neighbor can be in $X'$ or $X''$ and we do not gain the advantages of partitioning the space.

As we grow $q$ we approach a strong triangle inequality comparison in a strong metric space. In the limit – take q-square roots on both sides of (6A) and (6B) to obtain maxima –, the inequalities in (6A) and (6B) become

$$\mu < \max(d(u, x_o), d(x_o, v)). \quad \text{(7A)} \qquad d(x_o, v) \leq \max(\mu, d(u, x_o)). \quad \text{(7A)}$$

If we are searching in an infinity-metric (ultrametric) space we can still argue that: At least one of (7A) or (7B) must hold. If (7A) does *not* hold, the nearest neighbor must be in $X'$. If (7B) does *not* hold, the nearest neighbor must be in $X''$. However, and different from (5) and (6A)-(6B), we show in Lemma 1 of Appendix Appendix C that it is impossible for (7A) and (7B) to hold simultaneously. I.e., one and exactly one of (7A) or (7B) holds. From this lemma it follows that comparisons in ultrametric spaces always partition the dataset and we can therefore derive the following theorem.

**Theorem 1.** *Consider a dataset $X$ with $m$ elements and a dissimilarity function $d$ satisfying the strong triangle inequality* (3). *There is a search algorithm such that the number of comparisons $c(x_o)$ required to find the nearest neighbor $\hat{x}_o \in X$ of any query $x_o$ is bounded as*

$$c(x_o) \leq \left\lceil \log_2 m \right\rceil. \tag{6}$$

The proof of Theorem 1 is constructive (Appendix C). Search with a vantage point tree attains (6).

It follows from Theorem 1 that, given a choice, we would like to solve nearest neighbor search in ultrametric spaces. Alas, dissimilarity metrics of interest do not satisfy the strong triangle inequality. Some are not even metric. Due to this mismatch, we propose here an approach to *approximate* nearest neighbor search based on the development of projection and embedding operators to compute and approximate $q$-metric distances:

**Projection Operator.** The projection operator $P_q$ maps dissimilarities $d(x, x') \in D$ that do not necessarily satisfy the $q$-triangle inequality into distances $d_q(x, x') \in D_q$ that do satisfy the $q$-triangle inequality (Section 3).

**Embedding Operator.** The embedding operator $\Phi_q$ is a learned map from vectors $x \in \mathbb{R}^n$ into vectors $x_q \in \mathbb{R}^s$ such that 2-norms $\|x_q - x'_q\|_2$ approximate $q$-distances $d_q(x, x')$. The map is trained on the dataset $X$ and applied to queries $x_o$ (Section 4).

We use the projection operator $P_q$ to process the dataset $G = (X, D)$ to produce a graph $G_q = (X, D_q)$ such that any three points $x, y, z \in X$ satisfy the $q$-triangle inequality. This is a one-time preprocessing cost that we leverage for all queries. We use the learned embedding operator $\Phi_q$ to

approximate the $q$-metric distances between a query $x_o$ and points $x \in X$ as $d_q(x_o, x) \approx \|x_{oq} - x_q\|_2$ with $x_{oq} = \Phi_q(x_o)$. This is needed because computing the true $q$-metric distance $d_q(x_o, x)$ requires comparing $x_o$ to all points in $X$ (Section 3). Approximating $q$-metric distances with the embedding operator $E_q$ renders this search methodology approximate. Numerical experiments show that we are competitive with state-of-the-art approximate search methods (Section 5).

## 3 PROJECTION OF DISSIMILARITY FUNCTIONS ON $q$-METRIC SPACES

We seek to design a *q-metric projection* $P_q$ : $(X, D) \rightarrow (X, D_q)$ where, for arbitrary input dissimilarities $D$, the distances in $D_q$ satisfy the $q$-triangle inequality given in (4). Observe that finding a feasible projection is trivial. For example, consider a projection that assigns all distances in $D_q$ to be 1, regardless of the input $D$. In this case, the image space $(X, D_q)$ becomes an ultrametric (hence, a valid $q$-metric for all $q$). However, performing nearest neighbor search in $(X, D_q)$ would be a poor proxy for searching in $(X, D)$, as all distance information was lost in the projection. Motivated by this example, we build on the work of Segarra et al. (2015), which imposes conditions on $P_q$ to ensure that meaningful distance information is preserved through the projection. In their work, these requirements were formalized as the following two axioms:

**(A1) Axiom of Projection.** The $q$-metric graph $G_q = (X, D_q)$ is a fixed point of the projection map $P_q$, i.e.,

$$P_q(G_q) = G_q. \tag{A1}$$

**(A2) Axiom of Transformation.** Consider any two graphs $G = (X, D)$ and $G' = (X', D')$ and a dissimilarity reducing map $\varphi : X \rightarrow X'$ such that $D(x, y) \geq D'(\varphi(x), \varphi(y))$ for all $x, y \in X$. Then, the output $q$-metric graphs $(X, D_q) = P_q(G)$ and $(X', D'_q) = P_q(G')$ satisfy, for all $x, y \in X$,

$$D_q(x, y) \geq D'_q(\varphi(x), \varphi(y)). \tag{A2}$$

Axiom (A1) is natural in our setting: if the graph under study already satisfies the $q$-triangle inequality, then the projection should introduce no distortion and simply return the same graph. Axiom (A2) enforces a notion of monotonicity: if the distances in one graph dominate those in another, this dominance should be preserved under projection. Though seemingly lax, Axioms (A1) and (A2) impose significant structure on the projection $P_q$. In fact, we leverage the fact that *there exists a unique projection $P_q$ that satisfies Axioms (A1) and (A2)* Segarra et al. (2015). Moreover, we show that *this projection preserves nearest neighbors*. To formally state these results, we first introduce the canonical $q$-metric projection.

**Canonical $q$-metric projection $P_q^\star$.** Consider a graph equipped with a dissimilarity $G = (X, D)$ and let $C_{xy}$ denote the set of all paths from $x$ to $y$, where $x, y \in X$. For a given path $c = [x = x_0, x_1, \ldots, x_l = y] \in C_{xy}$, we define its $q$-length as $\ell_q(c) = \|[d(x, x_1), \ldots, d(x_{l-1}, y)]\|_q$. That is, the $q$-length of a path is the $q$-norm of the vector containing the dissimilarities along the path's edges. We define the canonical $q$-metric projection $(X, D_q) = P_q^\star(X, D)$ as

$$d_q(x, y) = \min_{c \in C_{xy}} \ell_q(c). \tag{7}$$

In words, the canonical $q$-metric projection computes the all-pairs shortest paths using the $q$-norm as the path cost function.

A first observation is that $P_q^\star$ is indeed a valid $q$-metric projection; i.e., that the output graph is guaranteed to satisfy the $q$-triangle inequality (see Appendix D). More importantly, as it was already proved in Segarra *et al.*'s work Segarra et al. (2015), $P_q^\star$ is uniquely characterized by the axioms of projection and transformation.

**Theorem 2** (Existence and uniqueness)**.** *The canonical projection $P_q^\star$ satisfies Axioms (A1) and (A2). Moreover, if a $q$-metric projection $P_q$ satisfies Axioms (A1) and (A2), it must be that $P_q = P_q^\star$.*

As we discuss next, Theorem 2 has direct practical applications for nearest-neighbor search.

### 3.1 SEARCH IN PROJECTED $q$-METRIC SPACES

Since the Axiom of Transformation (A2) is satisfied, we expect $P_q^\star$ to preserve up to some extent the distance ordering of the original graph, but this was never proved. In particular, we now show that the

nearest neighbors are preserved by this projection. To formalize this, consider the graph $H = (Y, E)$ whose node set $Y = X \cup \{x_o\}$ includes the dataset $X$ and the query $x_o$. The dissimilarity set $E$ consists of the original dissimilarities $D$ as well as the dissimilarities between the query and each data point, i.e., $E(x_o, x) = d(x_o, x)$ for all $x \in X$. We now apply the canonical projection to this extended graph, obtaining $(Y, E_q) = P_q^\star(H)$, where $E_q$ includes projected distances of the form $E_q(x, x_o)$ between each data point $x \in X$ and the query $x_o$. We next state that this projection preserves the identity of the nearest neighbor of $x_o$.

**Proposition 1.** *Given a graph $G = (X, D)$ and a query $x_o$, define $H = (Y, E)$ with $Y = X \cup \{x_o\}$ and let $(Y, E_q) = P_q^\star(H)$ be the projected graph. Then, the set of nearest neighbors satisfies*

$$\hat{x}_o \equiv \operatorname*{argmin}_{x \in X} E(x, x_o) \subseteq \operatorname*{argmin}_{x \in X} E_q(x, x_o). \tag{8}$$

*Moreover, for the case where $q < \infty$, the result holds with an equality.*

Proposition 1 shows that the canonical projection $P_q^\star$ in (7) preserves the nearest neighbor. This is enticing because we have argued that search in $q$-metric spaces is easier (Figure 1) and proved that search in $\infty$-metric spaces requires a logarithmic number of comparisons (Theorem 1). However, Equation (8) does not immediately help us solve the original nearest neighbor problem in (1). The proposition is stated for the projected dissimilarity set $E_q$, which requires access to the full set $E$, including all dissimilarities $E(x_o, x) = d(x_o, x)$ for every $x \in X$. Thus, computing $E_q$ requires comparing $x_o$ with all points in $X$, which defeats the purpose of reducing the number of comparisons $c(x_o)$. A broader analysis of this projection complexity is available in Appendix D.3. In the following section, we address the projection cost challenge by learning an embedding operator that enables us to *approximate* the values $E_q(x, x_o)$ without computing all pairwise distances explicitly.

## 4 Embedding Operator: Learning to Approximate $q$-metrics

We approximate $q$-metrics $E_q(x, x_o)$ with the 2-norm of a parameterized embedding $\Phi(x; \theta)$. Formally, we want to find a function $\Phi(x; \theta) : \mathbb{R}^n \to \mathbb{R}^s$ such that for any $x_o \in \mathbb{R}^n$ and $x \in X$:

$$E_q(x, x_o) = \big\| \Phi(x; \theta) - \Phi(x_o; \theta) \big\| \tag{9}$$

where $\| \cdot \|$ denotes the 2-norm of a vector. We fit the parameter $\theta$ to approximate distances $D_q(x, y)$ given by the canonical projection $D_q = P_q^\star(D)$, by minimizing a quadratic loss,

$$\ell_{\mathrm{D}}(x, y) = \Big[ D_q(x, y) - \big\| \Phi(x; \theta) - \Phi(y; \theta) \big\| \Big]^2. \tag{10}$$

We also consider an additional loss that measures the extent to which the $q$-triangle inequality is violated by the embedded distances. We choose a saturated linear penalty for this loss, explicitly,

$$\ell_{\mathrm{T}}(x, y, z) = \Big[ \big\| \Phi(x; \theta) - \Phi(y; \theta) \big\|^q - \big\| \Phi(x; \theta) - \Phi(z; \theta) \big\|^q - \big\| \Phi(y; \theta) - \Phi(z; \theta) \big\|^q \Big]_+. \tag{11}$$

where $[\cdot]_+$ denotes the projection to the non-negative orthant. The loss $\ell_{\mathrm{T}}$ is positive only when the $q$-triangle inequality is violated, in which case it takes on the value of the violation.

We minimize a linear combination of the losses $\ell_{\mathrm{D}}(x, y)$ and $\ell_{\mathrm{T}}(x, y, z)$ averaged over the dataset $X$

$$\theta^\star = \operatorname*{argmin}_{\theta} \alpha_{\mathrm{D}} \sum_{x, y \in X} \ell_{\mathrm{D}}(x, y) + \alpha_{\mathrm{T}} \sum_{x, y, z \in X} \ell_{\mathrm{T}}(x, y, z) \tag{12}$$

Notice that in (12) the loss $\ell_{\mathrm{T}}$ is redundant. It encourages the Euclidean norm $\big\| \Phi(x; \theta) - \Phi(y; \theta) \big\|$ to satisfy the $q$-triangle inequality when this is already implicit in $\ell_{\mathrm{D}}$ as the latter encourages proximity to distances $D_q(x, y)$ that we know satisfy the $q$-triangle inequality. We have observed in numerical experiments that adding $\ell_{\mathrm{T}}$ improves performance (see Appendix E). Further notice that although we train $\theta$ over the dataset $X$ we expect it to generalize to estimate the $q$-metric $E_q(x, x_o)$. Our numerical experiments in the next section indicate that this generalization is indeed successful.

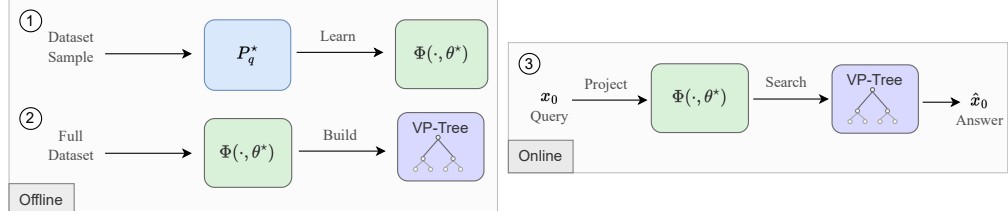

Figure 2: Infinity Search pipeline. Offline (left): Dataset samples are projected into a $q$-metric space using the canonical projection, and used to fit an embedding projection (1). Then, the dataset is transformed with the learned projection to build the VP-tree (2). Online (right): A query $x_o$ is transformed with the learned projection and then search is conducted using the VP-tree index.

## 5 EXPERIMENTS

We validate the properties of the canonical projection $E_q$ as well as the effects of the learned approximation $\Phi(\cdot, \theta^\star)$ in practical vector search settings. Since projecting the data with $P_q^\star$ is computationally expensive, we randomly sample a smaller subset of 1,000 points to evaluate the theoretical properties. To analyze the learned map $\Phi(\cdot, \theta)$, we use 10,000 samples from the Fashion-MNIST (Xiao et al., 2017) dataset. In all approximation experiments, we parameterize the embedding projector $\Phi$ using the same fully connected Multi-Layer Perceptron (MLP) architecture. The full dataset $X$ is projected using the trained MLP, and a VP-tree index is built on the resulting embedding. The Infinity Search framework is summarized in Figure 2. Details about the neural network architecture, training hyperparameters, and additional experimental results—including settings such as searching for $k \in 5, 10$ nearest neighbors or more datasets— are provided in Appendix E.

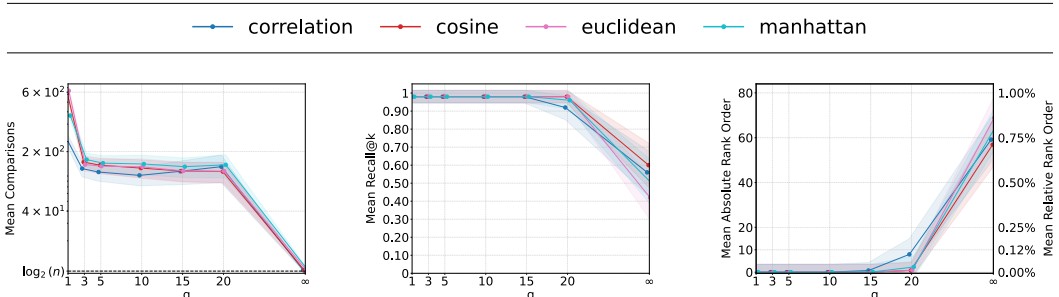

Figure 3: Number of comparisons and rank order when searching over MNIST-Fashion-784 ($k = 1$) with Canonical Projection $E_q$ for $n = 1,000$ points. Solid lines denote the mean and shading the standard deviation across queries.

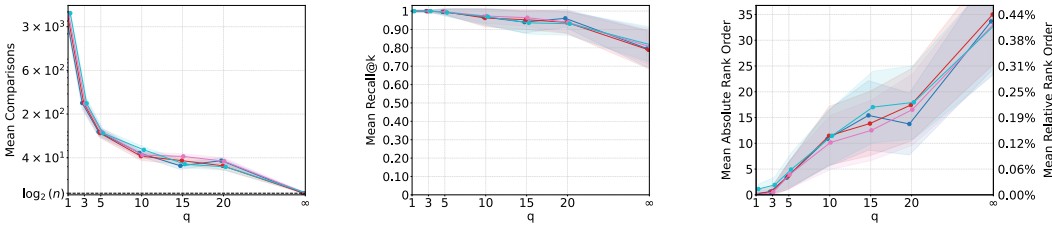

Figure 4: Number of comparisons and rank order when searching over MNIST-Fashion-784 ($k = 1$) with learned embedding $\Phi(\cdot, \theta^\star)$ for $n = 10,000$. Solid lines denote the mean and shading the standard deviation across queries.

**Searching with the canonical projection.** To analyze the effect of the $q$-metric structure on search complexity—decoupling it from that of the learned projection—we conduct experiments searching in the transformed graph $G_q$. We utilize a subset of the data due to the high computational cost

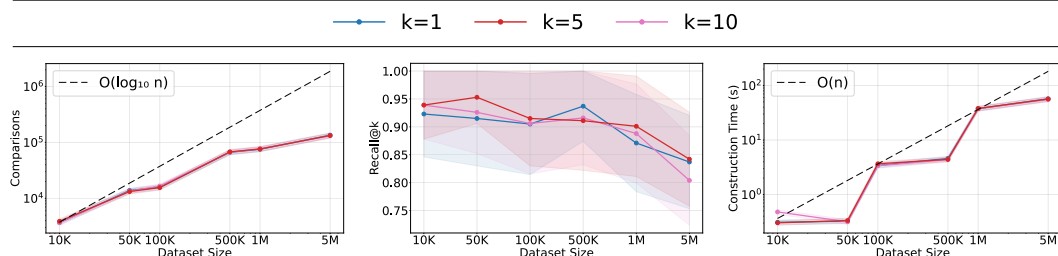

Figure 5: Infinity Search results on searching $n \in \{10\text{K}, 50\text{K}, 100\text{K}, 500\text{K}, 1\text{M}, 5\text{M}\}$ points of Deep1B-96 with Euclidean distance for $k \in \{1, 5, 10\}$.

associated with the canonical projection $P_q^\star$. As shown in Figure 3, increasing $q$ leads to a smooth and significant decrease in the number of comparisons. The boundary stated in 1 is reached for $q = \infty$, effectively confirming claim C1. The fact that the projection consistently returns the nearest neighbor, further supports the satisfaction of the $q$-triangle inequality (2). In addition, the relative error shown in the middle column of Figure 3 indicates preservation of the nearest neighbor, described in claim C2 and proved in 8. At $q = \infty$, the projection may introduce spurious optima not present in the original nearest neighbor set, thereby affecting accuracy. A similar effect occurs for large $q$, as distances between points can become artificially close, imitating this behavior.

**Learning approximation error.** If the learning process attains sufficiently low generalization error, the theoretical properties are expected to hold in practice. Figure 4 shows a reduction in the number of comparisons as $q$ increases, mirroring the trend observed in the Canonical Projection experiments. This reduction is monotonic with $q$, consistent across dissimilarities, and is accompanied by a moderate increase in rank error and a drop in recall. Recall values above 0.9 still yield speedups of up to three orders of magnitude. In all cases, the rank order remains below $1\%$ of the indexed dataset. The trend observed in absolute rank order reflects partial locality preservation by the learned embedding. A deeper analysis of the learning component–specifically, the projection's ability to fit and generalize–is deferred to Appendix E.

These approximation experiments support the satisfactory fulfillment of claim C3, which concerns the generalization of the learned projection to unseen data. This is supported by the observed preservation of speedup and a generalization error that remains low—reflected in high recall—relative to what the theoretical properties would predict.

**Scaling with the number of points.** While the current implementation is not yet optimized for large $n$, our analysis shows no inherent complexity barrier to scaling. Figure 5 (Deep1B subsets (Babenko and Lempitsky, 2016)) shows empirically sub-logarithmic growth in the number of comparisons with $n$. Search complexity is competitive with strong baselines: HNSW (Malkov and Yashunin, 2020) behaves as $\mathcal{O}(\log n)$ for fixed ef, IVF–PQ (Jégou et al., 2011) achieves sublinear $\mathcal{O}(n^\alpha)$ behavior, and ScaNN (Guo et al., 2020b) attains sublinear latencies via pruning with vector quantization. Build costs are likewise competitive: $\mathcal{O}(n)$ for Infinity Search, $\mathcal{O}(n \log n)$ for HNSW, $\mathcal{O}(n)$ for IVF–PQ after codebook training, and near-linear preprocessing for ScaNN. The projection $P_q^*$ is trained once on 100K points and then applied inductively with a fixed per-point projection cost across larger corpora. As expected with a fixed training size, accuracy declines as $n$ grows, but Recall@$k$ remains within a satisfactory range (see Fig. 5). For a more detailed analysis on scaling, see Appendix E.6.

## 5.1 INFINITY SEARCH IS COMPETITIVE IN ANN-BENCHMARKS

In this section, our end-to-end approach is compared against modern and classical vector search algorithms, such as HNSW (Malkov and Yashunin, 2018), IVF-PQ (Jégou et al., 2011) implemented in FAISS (Douze et al., 2024) or more recent approaches like ScaNN (Guo et al., 2020b). We use ANN-Benchmarks (Aumüller et al., 2018), a popular evaluation framework for in-memory Approximate Nearest Neighbor algorithms. While exhaustive benchmarking on substantially larger corpora is outside our current scope given the computational cost of computing full Pareto fronts, we complement our evaluation with the scaling curves in Fig. 5, which characterize performance trends as data size grows.

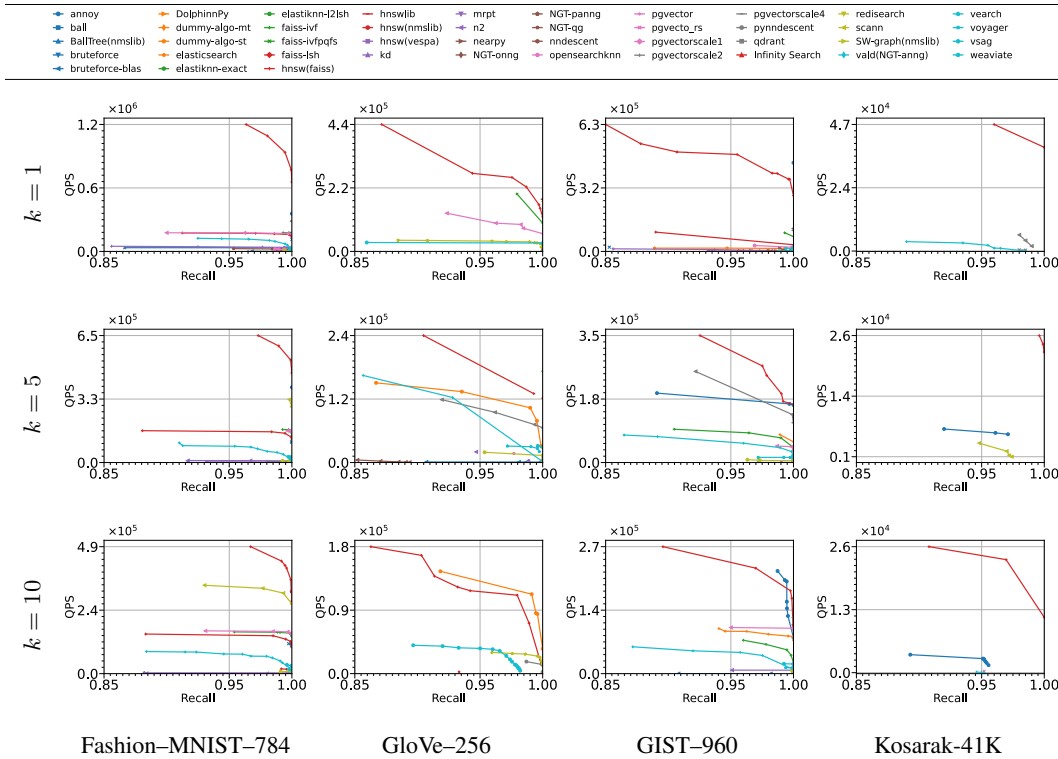

Figure 6: Search accuracy (Recall@$k$). Columns: datasets; rows: $k \in \{1, 5, 10\}$. Each panel: recall vs QPS across methods.

In this setting, search algorithms are evaluated based on their ability to trade off search accuracy and speed. Speed is measured by queries-per-second throughput. Note that the learning phase in our method is carried out offline during index building and that computing the embedding projection is a query pre-processing step. Search accuracy is measured using Rank Order for $k \in \{1, 5, 10\}$ – i.e. the position of the vector retrieved.

Across all datasets, our method either closely matches or outperforms baselines in nearest neighbor search for all values of $k$. Each point in the speed-accuracy curve in Figure 6 depicts a different hyperparameter setting by the benchmark. Notably, our method pareto dominates baselines by a large margin for settings with lower $k$. That is, considerable speed gains can be obtained, with minimal accuracy trade-offs, if higher order metric structure is imposed. In the case of searching for more neighbors ($k \in \{5, 10\}$), we still observe that our method performs competitively but with a less noticeable advantage. In particular, our method stands out on Kosarak, which is high-dimensional (41K), sparse, and evaluated with Jaccard similarity. This combination is unsupported or inefficient in many ANN libraries, making several baselines impractical. This highlights the ability of our method to accommodate arbitrary dissimilarities –like Cosine or Jaccard– and scale to higher dimensional data. We defer a more detailed analysis on experiments to Appendix E.7.

## 6 CONCLUSION

Driven by the insight that search in ultrametric spaces is logarithmic, in this paper we leverage a projection that endows a set of vectors with arbitrary dissimilarities with $q$-metric structure, while preserving the nearest neighbor. This projection of a dataset is computed as the shortest path in a graph using the $q$-norm as path length. To address the challenge of computing $q$-metric distances for queries efficiently, we developed a learning approach that embeds vectors into a space where Euclidean distances approximate $q$-metric distances. Our experiments real-world datasets, which included high-dimensional sparse data with less common dissimilarity functions, demonstrated that our method is competitive with state-of-the-art approximate search algorithms.

## REPRODUCIBILITY STATEMENT

We have taken several steps to ensure the reproducibility of our results. All theoretical contributions are accompanied by complete proofs in the appendix (see Appendices B–D for the $q$-metric projection properties and theorems). Experimental details, including network architectures, training hyperparameters, and additional results across multiple datasets and similarity measures, are provided in Appendix F. To facilitate benchmarking, we relied on the standardized `ANN-Benchmarks` framework Aumüller et al. (2018), ensuring fair and comparable evaluation with existing methods. In addition, we will release a supplementary `Docker` application containing the full training and evaluation pipeline, including data preprocessing, model training, and evaluation scripts. This will allow others to reproduce all reported figures and tables with minimal setup.

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

# A   RELATED WORK

The literature on vector-based nearest–neighbour search is vast and spans many decades. We briefly review the families that are most germane to our contributions and direct the reader to existing surveys Echihabi et al. (2021); Ukey et al. (2023); Li et al. (2019) for a broader analysis.

For data sets endowed with a proper metric, pruning rules derived from the standard triangle inequality have long been exploited through pivot- or vantage-point trees, $k$-d trees, ball trees, and their variants Yianilos (1993); Uhlmann (1991); Beygelzimer et al. (2006). Although these methods can yield exact results, they typically struggle with dimensionality and rely on existing metric structure of the date have been largely ouperformed by inverted file indexes Babenko and Lempitsky (2014), hashing Indyk and Motwani (1998); Wang et al. (2014), quantisation Kalantidis et al. (2014); Ge et al. (2020) and graph based methods Malkov and Yashunin (2018) such as HNSW Malkov and Yashunin (2018). Approaches—such as DiskANN Dong et al. (2020), SPANN Yu et al. (2015), and ONNG Arai (2018)— have optimized memory layout, link degree, or storage hierarchy to scale to billion-point datasets.

While many of these methods can offer strong performance, recent works increasingly seek to embed learning into the indexing process for data-aware improvements. ScaNN Guo et al. (2020a) jointly optimizes clustering and quantization, SOAR adds orthogonality-amplified residuals to further boost ScaNN's efficiency Sun et al. (2023), and AutoIndex Zhao et al. (2021) automates the selection of IVF–PQ parameters. Learned policies can also steer the search itself: L2-KNN Cao et al. (2021) and LARK Cerda et al. (2024) guide edge traversal at search time in graph based methods to reduce distance computations, AdaptNN predicts early-termination thresholds online Li et al. (2020), and Tao learns to set them *before* query execution using only static local-intrinsic-dimension features Yang et al. (2021). Differentiable hierarchies, such as Hierarchical Quantized Autoencoders Roy et al. (2023) and LION Raguso et al. (2024), train the entire coarse-to-fine quantization pipeline end-to-end. LoRANN replaces PQ scoring with a supervised low-rank regressor that outperforms traditional IVF on billion-scale data Jääsaari et al. (2024).

Industrial adoption of large language models has driven the emergence of "vector DBMSs", see Pan et al. (2024) and references therein. Despite many developments in compression, partitioning, graph navigation, and hybrid attribute-vector query planning, most if not all systems index the *original* dissimilarities - or an approximation – and thus inherit their metric limitations. In contrast, we propose to pre-process data to obtain a $q$-metric space whose structure is provably more favourable for search.

# B   GENERALIZED VANTAGE POINT TREE

In this section, we modify the classical VP-tree Hanov (2011). The standard index prunes subtrees using the triangle inequality, yielding exact search with fewer distance evaluations. In high dimensions, however, concentration of measure makes pairwise distances cluster, so these tests rarely trigger and the VP-tree approaches brute-force search. We address this by generalizing the pruning scheme to $q$-metric spaces. Let $D_q$ denote a dissimilarity that satisfies the $q$-triangle inequality

$$d(x, z)^q \le d(x, y)^q + d(y, z)^q \tag{13}$$

As $q$ increases, (13) tightens, which strengthens the pruning tests and typically reduces distance evaluations even in higher dimensions. In the limit $q \to \infty$, (13) converges to the strong triangle inequality:

$$d(x, z) \le \max\{d(x, y), d(y, z)\} \tag{14}$$

leading to even stronger pruning conditions. Moreover, as described in Appendix B.2, if $q$-Metric structure is present in the data, the modified tree preserves the Nearest Neighbor.

## B.1  $q$-VP-TREE

The $q$-VP-Tree is a binary tree constructed by recursively partitioning the set of points $X$ based on its pairwise distances. As detailed in Algorithm 1, begin by randomly selecting a root vantage point $v_{\text{root}} \in X$ and setting

$$\mu_{v_{\text{root}}} := \text{median}\big\{ \, d(x, v_{\text{root}}) : x \in X \setminus \{v_{\text{root}}\} \, \big\}. \tag{15}$$

This induces the disjoint children

$$X' := \{ \, x \in X \setminus \{v_{\text{root}}\} : d(x, v_{\text{root}}) \leq \mu_{v_{\text{root}}} \, \}, \qquad X'' := \{ \, x \in X \setminus \{v_{\text{root}}\} : d(x, v_{\text{root}}) > \mu_{v_{\text{root}}} \, \}, \tag{16}$$

with $X' \cap X'' = \varnothing$ and $X' \cup X'' = X \setminus \{v_{\text{root}}\}$. Because the split is taken at the median of the $|X| - 1$ distances, the partitions also satisfy

$$\max \left[ |X'|, |X''| \right] = \left\lfloor \frac{|X|}{2} \right\rfloor. \tag{17}$$

The same procedure is applied recursively to $X'$ and $X''$ to obtain the left and right subtrees, until no partition is possible, yielding a total of $h = \lceil log_2(|X|) \rceil$ levels in the tree.

After the tree is constructed, given a query point $x_o$ it can be traversed as described in Algorithm 2. Begin at root node $v_{root}$ and threshold $\mu_{v_{root}}$ and set $\tau = \min\{\tau, d(x_o, v_{root})\}$ as the minimum distance found so far. The $q$-triangle inequality enforces $d(x_o, v_{root})^q \leq \tau^q + \mu_{v_{root}}^q$ to hold in the left child and $d(x_o, v_{root})^q \geq \mu_{v_{root}}^q - \tau^q$ in the right child. Consequently, if neither condition holds both children are visited because neither bound rules out a closer point. In the presence of ultrametric structure where $q = \infty$, Lemma 1 yields a disjoint partition: if $\max \left[ d(x_o, v_{root}), \tau \max \right] \leq \mu_{v_{root}}$ the right child is safely pruned, otherwise the left child is pruned. Since this process has to be repeated recursively, Algorithm 3 explores at most one branch per node at $q = \infty$.

---

**Algorithm 1** Construction Phase of $q$-VPTree

---

1: **function** BUILDVPTREE($X$)
2:     Select a random vantage point $v \in X$
3:     Let $D \leftarrow \{d(x, v) \mid x \in X \setminus \{v\}\}$
4:     Let $\mu \leftarrow$ median of $D$
5:     Partition $X \setminus \{v\}$ into:
        $X' \leftarrow \{x \in X \setminus \{v\} : d(x, v) \leq \mu\}$
        $X'' \leftarrow \{x \in X \setminus \{v\} : d(x, v) > \mu\}$
6:     left $\leftarrow$ BUILDVPTREE($X'$)
7:     right $\leftarrow$ BUILDVPTREE($X''$)
8:     **return** $[v, \text{left}, \text{right}, \mu]$
9: **end function**

---

Despite its advantages, the $q$-VP Tree relies on the assumption of an underlying $q$-metric space, thus limiting its applicability. The Projection exposed in Section 3, allows enforcing this structure at any data, endowing this algorithm with unrestricted use.

**Algorithm 2** Searching Phase of the $q$-VPTree

1: **function** SEARCHVPTREE($x_o, q, v, \tau$)
2:     $\tau = \min(\tau, d(x_o, v))$
3:     **if** $d(x_o, v)^q \leq \mu_v^q - \tau^q$ **then**
4:         nn $\leftarrow$ SEARCHVPTREE($x_o, q, v.\text{left}, \tau$)
5:     **end if**
6:     **if** $d(x_o, v)^q > \mu_v^q + \tau^q$ **then**
7:         nn $\leftarrow$ SEARCHVPTREE($x_o, q, v.\text{right}, \tau$)
8:     **end if**
9:     **if** $\mu_v^q - \tau^q < d(x_o, v)^q \leq \mu_v^q + \tau^q$ **then**
10:       nn $\leftarrow$ SEARCHVPTREE($x_o, q, v.\text{left}, \tau$)
11:       nn $\leftarrow$ SEARCHVPTREE($x_o, q, v.\text{right}, \tau$)
12:     **end if**
13:     **return** nn
14: **end function**

---

**Algorithm 3** Searching Phase of the $\infty$-VPTree

1: **function** SEARCHVPTREE($x_o, q, v, \tau$)
2:     $\tau = \min(\tau, d(x_o, v))$
3:     **if** $\max\left[d(x_o, v), \tau\right] \leq \mu_v$ **then**
4:         nn $\leftarrow$ SEARCHVPTREE($v.\text{left}, x_o, \tau, q$)
5:     **end if**
6:     **if** $max\left[\mu_v, \tau\right] < d(x_o, v)$ **then**
8:         nn $\leftarrow$ SEARCHVPTREE($v.\text{right}, x_o, \tau, q$)
9:     **end if**
10:     **return** nn
11: **end function**

### B.2   $q$-VPTREE PRESERVES THE NEAREST NEIGHBOR

We follow the $q$-VPTree construction in Segarra et al. (2015).

**Proposition 2.** *When using a VP-tree for nearest neighbor search in a $q$-metric space, the following pruning rules ensure that no potentially optimal node is discarded:*

$$\begin{cases} i) & D_q^q(x_o, v) \leq \mu_v^q - \tau^q & \Rightarrow \textit{visit only the left child,} \\ ii) & \mu_v^q - \tau^q < D_q^q(x_o, v) \leq \mu_v^q + \tau^q & \Rightarrow \textit{visit both children,} \\ iii) & \mu_v^q + \tau^q < D_q^q(x_o, v) & \Rightarrow \textit{visit only the right child,} \end{cases} \tag{18}$$

*where $x_o$ is the query point, $v$ is the vantage point at the current node of the VP-tree, $\mu_v$ is the median distance at that node, $\tau$ is the current best-so-far distance, and $D_q$ are the pairwise distances between indexed points living in a $q$-metric space.*

*Proof.* We show that applying the rules in (18) ensures that no candidate point closer than $\tau$ to the query point $x_o$ is discarded.

**Case (i):** Assume $D_q^q(x_o, v) \leq \mu_v^q - \tau^q$. Then, by the $q$-triangle inequality, for any point $t \in X$,

$$D_q^q(v, t) \leq D_q^q(v, x_o) + D_q^q(x_o, t) \leq \mu_v^q - \tau^q + D_q^q(x_o, t). \tag{19}$$

For every $t$ in the right child, we know by construction that $D_q(v, t) > \mu_v$, so $D_q^q(v, t) > \mu_v^q$. Combining this with (19), we get:

$$\mu_v^q < \mu_v^q - \tau^q + D_q^q(x_o, t) \quad \Rightarrow \quad D_q^q(x_o, t) > \tau^q, \tag{20}$$

implying $D_q(x_o, t) > \tau$. Thus, all points in the right child are farther from $x_o$ than the current best distance and can be safely pruned.

**Case (iii):** Assume $\mu_v^q + \tau^q < D_q^q(x_o, v)$. Using the $q$-triangle inequality again, for any $t \in X$,

$$D_q^q(x_o, v) \leq D_q^q(x_o, t) + D_q^q(t, v). \tag{21}$$

Since all points $t$ in the left child satisfy $D_q(t, v) \leq \mu_v$, we have $D_q^q(t, v) \leq \mu_v^q$. Combining with (21):

$$\mu_v^q + \tau^q < D_q^q(x_o, v) \leq D_q^q(x_o, t) + \mu_v^q \quad \Rightarrow \quad D_q^q(x_o, t) > \tau^q, \tag{22}$$

and hence $D_q(x_o, t) > \tau$, so the left child can be pruned.

**Case (ii):** This is the ambiguous region where neither child can be safely discarded based solely on the bounds, so both must be visited to guarantee correctness.

This concludes the proof. $\qquad \square$

## C  NEAREST NEIGHBOR SEARCH AND METRIC STRUCTURE

The explanations presented in Section C demonstrate that, under the metric structure defined by equation 4, the complexity of nearest neighbor search can be reduced. In particular, for the limit case where $q = \infty$, such reduction is guaranteed by the *strong triangle inequality*:

$$d(x, y) \leq \max\Big[d(x, z), d(y, z)\Big]. \tag{23}$$

This guarantee follows from the metric structure: at each vantage point it induces a mutually exclusive partition of points that allows safely discarding candidate neighbors. This partition is reflected in following lemma:

**Lemma 1.** *If a symmetric dissimilarity function $d$ satisfies the strong triangle inequality for any three points $x_o, u, v \in \mathbb{R}^n$, then for any $\mu > 0$, one and exactly one of the following conditions holds*

$$\max\Big[d(u, x_o), d(x_o, v)\Big] \leq \mu \tag{24}$$

$$\max\Big[\mu, d(u, x_o)\Big] < d(x_o, v) \tag{25}$$

*Proof.* The proof will follow by contradiction. Assume (24) and (25) hold, then by the strong triangle inequality we have $d(u, v) \leq \max\Big[d(u, x_o), d(x_o, v)\Big] \leq \mu$. Therefore, it follows

$$d(x_o, v) \leq \max\Big[d(v, u), d(x_o, v)\Big] \leq \max\Big[\mu, d(u, x_o)\Big] < d(x_o, v) \tag{26}$$

which yields to a contradiction. $\qquad \square$

This dichotomy generated by the strong triangle inequality, can be further exploited to purge candidate points. When combined with the modification of the classical VP-Tree presented in Appendix B, this dichotomy gives place to Theorem 1 of Section 2.

**Theorem 3.** *Consider a dataset $X$ with $m$ elements and a dissimilarity function $d$ satisfying the strong triangle inequality (3). There is a search algorithm such that the number of comparisons $c(x_o)$ required to find the nearest neighbor $\hat{x}_o \in X$ of any query $x_o$ is bounded as*

$$c(x_o) \leq \Big\lceil \log_2 m \Big\rceil. \tag{27}$$

*Proof.* Consider a VP-tree as defined in Appendix B, constructed on $X$ using Algorithm 1. For a query point $x_o$, it suffices to traverse the tree with Algorithm 3. By Lemma 1, the search makes one comparison per level; hence the total number of explored levels is at most

$$h = \Big\lceil \log_2 |X| \Big\rceil. \tag{28}$$

$\qquad \square$

This Theorem motivates the pursuing of metric structure on data. At $q = \infty$, logarithmic complexity can be attained. Moreover, as it will be empirically proved in Appendix 10a, this complexity reduction happens monotonically when $q$ is taken to $\infty$. Since $q$-metric structure does not always hold, it has to be imposed by means of the projection presented in Section 3.

# D  PROJECTION

In this Appendix, we expand on the theoretical properties exposed in Section 3. The proofs rely on the Axioms A1 and A2, that are imposed on the Canonical Projection. These two requirements are enough to imbue the projection with beneficial properties on Nearest Neighbor search.

## D.1  $P_q^\star$ EXISTS, IS UNIQUE, AND SATISFIES THE $q$-TRIANGLE INEQUALITY

The results in Section D.1 were presented in Segarra et al. (2015) and are included here for completeness.

For a given $q$, the Canonical Projection $P_q^\star$ as defined in 7, is guaranteed to satisfy the $q$-triangle inequality. Moreover, it meets axioms of Projection A1 and Transformation A2.

**Lemma 2** (Satisfaction of $q$-triangle inequality). *The canonical Projection as defined in 7, satisfies the q-triangle inequality.*

*Proof.* To verify the $q$-triangle inequality, let $c_{xx'}$ and $c_{x'x''}$ be paths that achieve the minimum in 7 for $d_q(x, x')$ and $d_q(x', x'')$, respectively. Then, from the definition in 7, it follows that:

$$d_q(x, x'')^q = \min_{c_{xx''}} \|c\|_q^q \leq \|c_{xx'} \oplus c_{x'x''}\|_q^q = \|c_{xx'}\|_q^q + \|c_{x'x''}\|_q^q = d_q(x, x')^q + d_q(x', x'')^q, \tag{29}$$

where the inequality holds because the concatenated path $c_{xx'} \oplus c_{x'x''}$ is a valid (though not necessarily optimal) path between $x$ and $x''$, while $d_q(x, x'')$ minimizes the $q$-norm across all such paths. $\square$

**Theorem 1** (Existence and uniqueness). The canonical projection $P_q^\star$ satisfies Axioms A1 and A2. Moreover, if a $q$-metric projection $P_q$ satisfies Axioms A1 and A2, it must be that $P_q = P_q^\star$.

*Proof.* We first prove that $d_q$ is indeed a $q$-metric on the node space $X$. That $d_q(x, x') = d_q(x', x)$ follows from the fact that the original graph $G$ is symmetric, and that the norms $\|\cdot\|_q$ are symmetric in their arguments for all $q$. Moreover, $d_q(x, x') = 0$ if and only if $x = x'$, due to the positive definiteness of the $q$-norms.

To see that the Axiom of Projection A1 is satisfied, let $M = (X, d) \in \mathcal{M}_q$ be an arbitrary $q$-metric space, and denote $(X, d_q) = P_q^\star(M)$, the output of applying the canonical $q$-metric projection. For any pair $x, x' \in X$, we have:

$$d_q(x, x') = \min_{c \in C_{xx'}} \|c\|_q \leq \|[x, x']\|_q = d(x, x'), \tag{30}$$

for all $q$, where the inequality comes from choosing the trivial path $[x, x']$ consisting of a single edge from $x$ to $x'$.

Let $c_{xx'}^\star = [x = x_0, x_1, \ldots, x_\ell = x']$ be the path that achieves the minimum in the above expression. Using Lemma 2 we know $d_q$ satisfies the $q$-triangle inequality:

$$d(x, x') \leq \left( \sum_{i=0}^{\ell-1} d(x_i, x_{i+1})^q \right)^{1/q} = \|c_{xx'}^\star\|_q = d_q(x, x'). \tag{31}$$

Substituting this into the previous inequality, we find that:

$$d(x, x') = d_q(x, x'). \tag{32}$$

Since $x$ and $x'$ were arbitrary, we conclude $d \equiv d_q$, hence $P_q^\star(M) = M$, as required.

To show that the Axiom of Transformation A2 holds, consider two graphs $G = (X, D)$ and $G' = (X', D')$, and a dissimilarity-reducing map $\varphi : X \to X'$. Let $(X, d_q) = P_q^\star(G)$ and $(X', d_q') = P_q^\star(G')$ be the projected spaces.

For a pair $x, x' \in X$, let $c_{xx'}^\star = [x = x_0, x_1, \ldots, x_\ell = x']$ be a path achieving the minimum for $d_q(x, x') = \|c_{xx'}^\star\|_q$. Consider the image path in $X'$:

$$P_{\varphi(x)\varphi(x')}^\star = [\varphi(x_0), \varphi(x_1), \ldots, \varphi(x_\ell)]. \tag{33}$$

Since $\varphi$ is dissimilarity-reducing, we have:

$$D'(\varphi(x_i), \varphi(x_{i+1})) \leq D(x_i, x_{i+1}) \quad \text{for all } i. \tag{34}$$

Hence,

$$\|P_{\varphi(x)\varphi(x')}^\star\|_q \leq \|c_{xx'}^\star\|_q = d_q(x, x'). \tag{35}$$

Now, since $d_q'(\varphi(x), \varphi(x'))$ is defined as the minimum over all such paths in $X'$, we get:

$$d_q'(\varphi(x), \varphi(x')) \leq \|P_{\varphi(x)\varphi(x')}^\star\|_q \leq d_q(x, x'), \tag{36}$$

which completes the proof of Axiom A2. $\qquad\qquad\qquad\qquad\qquad\qquad\qquad\qquad\square$

### D.2 CANONICAL PROJECTION PRESERVES THE NEAREST NEIGHBOR

As a consequence of the Axioms A1 and A2, the Canonical Projection $P_q^\star$ will preserve the Nearest Neighbor. This, when combined with Proposition 2

**Proposition 1.** Given a graph $G = (X, D)$ and a query $x_o$, define $H = (Y, E)$ with $Y = X \cup \{x_o\}$ and let $(Y, E_q) = P_q^\star(H)$ be the projected graph. Then, we have that the set of nearest neighbors satisfies

$$\hat{x}_o \equiv \operatorname*{argmin}_{x \in X} E(x, x_o) \subseteq \operatorname*{argmin}_{x \in X} E_q(x, x_o). \tag{37}$$

*Proof.* We prove the theorem by showing that any projection method that satisfies the Axiom of Transformation (A2) *cannot* result in larger dissimilarities for any pair of points. We further show that nearest neighbor dissimilarities are preserved by the canonical projection. The combination of these two facts yields (37).

To prove that dissimilarities are not increased, construct a graph $\tilde{G} = (\tilde{X}', \tilde{D}',)$ made up of two arbitrarily chosen nodes $x$ and $y$ along with their corresponding dissimilarities. I.e.,

$$\tilde{X} = \{x, y\}, \qquad \tilde{D} : \tilde{D}(x, y) = E(x, y). \tag{38}$$

Notice that, since $(X', E')$ is a metric space, any admissible projection will leave the space unchanged. This is a direct consequence of Axiom 1 in (A1).

Let us now map the original graph $G$ to the new one:

$$\varphi(y) = \begin{cases} x_1' & \text{if } x = x_o, \\ x_2' & \text{otherwise.} \end{cases} \tag{39}$$

Since the only existing edge has the minimum value possible $E(\hat{x}_o, x_o)$, $\varphi$ is can be considered dissimilarity-reducing map. As a consequence of Axiom 2 in (A2), since

$$E(x, x_o) \geq E(\hat{x}_o, x_o) = E'(x_1', x_2') \tag{40}$$

therefore, any admissible projection will satisfy:

$$E'(x, x_o) \geq E'(x_1', x_2') = E(\hat{x}_o, x_o) \tag{41}$$

This enforces any admissible map deliver distances greater than the one to the original Nearest Neighbor. Moreover, if we reason analogously but interchanging the roles of $x_o$ and its Nearest Neighbor, we can state that the distance is preserved:

$$E'(x, x_o) = E'(x_1', x_2')$$ (42)

By taking into account that the canonical projection is the only admissible map, since $E'$ must be satisfied by $E_q = P_q^\star(H)$. This, proofs the preservation of the optima:

$$\hat{x}_o \equiv \operatorname*{argmin}_{x \in X} E(x, x_o) \subseteq \operatorname*{argmin}_{x \in X} E_q(x, x_o).$$

□

Having a preservation of the nearest neighbors makes the projection a valid candidate for search. However, spurious neighbors could be added to the optimal set. This would incur in the cost of searching among non desired candidates. However, the previous result can be further extended with the following Lemma:

**Lemma 3.** *Under the setting of Proposition D.2, if $q < \infty$, the nearest neighbor is preserved with the following equality:*

$$\hat{x}_o \equiv \operatorname*{argmin}_{x \in X} E(x, x_o) = \operatorname*{argmin}_{x \in X} E_q(x, x_o).$$ (43)

*Proof.* The proof will follow by contradiction.

Assume there exists $z \in X$ such that

$$z \notin \operatorname*{argmin}_{x \in X} E(x, x_o) \quad \text{and} \quad z \in \operatorname*{argmin}_{x \in X} E_q(x, x_o)$$ (44)

We know by Theorem 2 that $E_q$ is generated by applying $P_q$, the canonical projection. Therefore there exists a path

$$c^* = [x_o, \dots, z] \quad \text{such that}$$ (45)

$$d_q(x_o, z) = ||c^*||_q = \min_{c \in \mathcal{C}_{x_o z}} \ell_q(c)$$ (46)

However, since $z \notin \operatorname{argmin}_{x \in X} E(x, x_o)$,
$[x_o, z]$ is not a valid minimum path, otherwise, this would imply

$$d(x_o, z) = \sqrt[q]{d(x_o, z)^q} = \ell_q([x_o, z]) = \min_{x \in X} d(x_o, x)$$ (47)

and consequently $z \in \operatorname{argmin}_{x \in X} E(x, x_o)$.

Without loss of generality, assume $\exists \, x_1 \neq z$ such that $c^* = [x_o, x_1, \dots, z]$ . Since $\ell_q$ is strictly increasing and by symmetry of dissimilarity,

$$\ell_q([x_o, x_1, \dots, z]) > \ell_q([x_o, x_1]) \geq \min_{c \in \mathcal{C}_{x_o z}} \ell_q(c)$$ (48)

$$\Rightarrow z \notin \operatorname*{argmin}_{x \in X} E_q(x, x_0)$$ (49)

Which incurs in a contradiction.

□

The preceding lemma ensures that no spurious neighbors are added, a property that is not necessarily satisfied by ultrametric spaces. The following example illustrates a case where the set of original set of nearest neighbors can be extended if $q = \infty$.

**Example 1.** Let $\{x_1, x_2, x_3\} = X$ be three points equipped with the original distances

$$D(x_1, x_2) = 3, \quad D(x_2, x_3) = 2, \quad D(x_1, x_3) = 5. \tag{50}$$

Clearly $D(x_1, x_2) < D(x_1, x_2)$, so the unique nearest neighbor of $x_1$ in $G = (\{x_1, x_2, x_3\}, D)$ is $x_2$. We now enforce the strong triangle inequality by projecting:

$$D'(x_2, x_3) = \max\Big[\{D(x_1, x_2), D(x_2, x_3)\}\Big] = 3, \tag{51}$$

while leaving $d'(x_1, x_2) = 3$ and $D'(x_2, x_3) = 2$. Under the new distance $D'$, one finds

$$D'(x_1, x_2) = 3 \quad \text{and} \quad D'(x_1, x_3) = 3, \tag{52}$$

so $x_1$ is equidistant from $x_2$ and $x_3$. In particular, the original nearest neighbor $x_2$ is no longer uniquely closest—any nearest-neighbor search on $G' = (\{x_1, x_2, x_3\}, D')$ may return $x_3$ instead of $x_2$, demonstrating that ultra-projecting a metric can extend original nearest neighbor set.

The proposition guarantees that the Nearest Neighbor is preserved under the mapping. Therefore, solving the solution found when solving Nearest Neighbor Search in the transformed space is a relaxation of the original problem. Although this relaxation provides speedup, it can also make the problem more difficult if many solutions are added to the nearest neighbor set. This problem is addressed in Section E, by further exploiting metric tree structure.

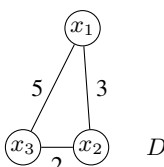

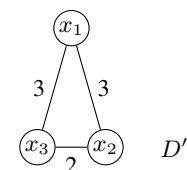

Figure 7: Ultrametric projection added nearest neighbors to the original set.

### D.3   EFFICIENT APPROXIMATION OF $P_q^*$

Calculating the canonical projection $P_q^\star$ over the training points can incur prohibitive computing times. Algorithms 4 and 5 describe the computation of the canonical projection. These procedures are highly paralellizable and can be efficiently executed on GPUs, which significantly reduces the projection time. Nevertheless, computing the canonical projection for $n$ points still requires $O(n^3)$ operations in the worst case, which remains prohibitive for large datasets.

Therefore, instead of computing the shortest $q$-norm path in the complete graph $D \in \mathbb{R}^{n \times n}$, we restrict the search to length $l$ paths and only consider the $k$-nearest neighbor graph. By doing this, the complexity of computing all pairwise distances is reduced from $O(n^3)$ to a factor $O(nk^2 l)$.

---

**Algorithm 4** Canonical Projection $P_q^*$

1: **function** $P^*(D, q)$
2:     $M \leftarrow D^q$
3:     **for** $k = 1$ to $n$ **do**
4:         $c_k \leftarrow M[:, k] + M[k, :]$
5:         $M \leftarrow \min(M, c_k)$
6:     **end for**
7:     **return** $M^{1/q}$
8: **end function**

**Algorithm 5** Canonical Projection $P_\infty^*$

1: **function** $P^*(D, \infty)$
2:     $M \leftarrow D$
3:     **for** $k = 1$ to $n$ **do**
4:         $c_k \leftarrow \max(M[:, k], M[k, :])$
5:         $M \leftarrow \min(M, c_k)$
6:     **end for**
7:     **return** $M$
8: **end function**

---

Figure 8: Algorithms for Canonical Projection computation. Complexity: $O(n^3)$

Given a dissimilarity distance matrix $D \in \mathbb{R}^{n \times n}$ (e.g., cosine or Euclidean), we raise it element-wise to the power $q$ to obtain edge weights consistent with the $q$-path metric. Then, for each node $x_i$, we consider only its $k$ nearest neighbors in the original space and perform a fixed number of smoothing updates, which approximate the optimal $q$-path cost to all other nodes.

At each iteration, for every node $x_i$ and each of its neighbors $x_j$, we update the path cost $D(x_i, x')$ to a third node $x'$ using the relaxed bound:

$$D(x_i, x') \leftarrow \min \left\{ D(x_i, x'), \ D(x_i, x_j)^q + D(x_j, x')^q \right\}. \tag{53}$$

The final result is root-transformed to return to the original distance scale:

$$d_q^*(x_i, x_j) \approx (D(x_i, x_j))^{1/q}. \tag{54}$$

These modifications are incorporated in Algorithms 6 and 7, where the adjacency matrix $A$ restricts the minimum-path computation to the $k$-nearest neighborhood. In addition, paths are truncated to length $l$, meaning that at most $l$ intermediate components can form a path. This approximated algorithm remains highly paralellizable in GPU while using sparse matrix multiplications.

---

**Algorithm 6** Sparse Canonical Projection $P_q^*$

1: **function** $P^*(D, A, l, q)$
2: $\quad M_{ij} \leftarrow \begin{cases} D_{ij}^q, & \text{if } A_{ij} = 1 \\ \infty, & \text{otherwise} \end{cases}$
3: $\quad$ **for** $k = 1$ to $l$ **do**
4: $\quad\quad c_k \leftarrow M[:, k] + M[k, :]$
5: $\quad\quad M \leftarrow \min(M, c_k)$
6: $\quad$ **end for**
7: $\quad$ **return** $M^{1/q}$
8: **end function**

**Algorithm 7** Sparse Canonical Projection $P_\infty^*$

1: **function** $P^*(D, A, l, \infty)$
2: $\quad M_{ij} \leftarrow \begin{cases} D_{ij}, & \text{if } A_{ij} = 1 \\ \infty, & \text{otherwise} \end{cases}$
3: $\quad$ **for** $k = 1$ to $l$ **do**
4: $\quad\quad c_k \leftarrow \max(M[:, k], M[k, :])$
5: $\quad\quad M \leftarrow \min(M, c_k)$
6: $\quad$ **end for**
7: $\quad$ **return** $M$
8: **end function**

Figure 9: Sparse Canonical Projection using adjacency $A$ (restricting updates to local neighbors) and early stopping after $l$ pivots (paths of length $\leq l + 1$). Complexity: $O(nk^2l)$.

# E  EXPERIMENTS & RESULTS

## E.1  EXPERIMENTAL SETTINGS

**Vector Datasets**

The experiments were conducted on two commonly used datasets and one with distinct characteristics. We split the data randomly, using 80% for indexing and the remaining 20% is used as queries. We provide a summary of the datasets below, additional details can be found in the references provided.

- **Xiao et al. (2017)Fashion-MNIST-784 Euclidean**: Image samples from the Fashion-MNIST collection, each flattened into a 784-dimensional vector.

- **Pennington et al. (2014)GloVe-200 Cosine**: Text embeddings extracted from the GloVe (Global Vectors for Word Representation) model, each of dimension 200.

- **Oliva and Torralba (2001)Gist-960 Euclidean**: A set of real-valued GIST descriptors of natural scene images, each represented as a 960-dimensional vector. A GIST descriptor is a global, low-dimensional representation of an image's spatial envelope.

- **Newman (2008)NYTimes-256 Cosine**: Document embeddings from the UCI NYTimes Bag-of-Words corpus. We convert term-count vectors to TF–IDF and apply truncated SVD to 256 dimensions; evaluation uses cosine (angular) distance.

- **Bodon (2003)Kosarak-41,000 Jaccard**: A real-world sparse binary transaction dataset derived from click-stream data of a Hungarian news portal. Each transaction is represented as a 41,000-dimensional vector.

- **Babenko and Lempitsky (2016)Deep1B-96 Euclidean**: A billion-scale collection of CNN-based image descriptors which are later PCA-reduced to 96 dimensions.

## Dissimilarities

Apart from searching using the euclidean distance, as customary for Glove and FashionMNIST, to showcase how our approach can accommodate arbitrary dissimilarity functions, we also evaluate our method and results searching with other dissimilarities on the same datasets. The dissimilarities used are specified in Table 1.

Table 1: Distance and Dissimilarity Metrics used.

| Metric | Formula |
|---|---|
| Euclidean Distance | $d(x,y) = \sqrt{\sum_{i=1}^{d}(x_i - y_i)^2}$ |
| Manhattan Distance | $d(x,y) = \sum_{i=1}^{d}|x_i - y_i|$ |
| Cosine Dissimilarity | $d(x,y) = 1 - \frac{x \cdot y}{\|x\| \, \|y\|}$ |
| Correlation | $d(x,y) = 1 - \frac{(x-\bar{x}) \cdot (y-\bar{y})}{\|x-\bar{x}\| \, \|y-\bar{y}\|}$ |

## Metrics

Approximate search algorithms are evaluated along two key dimensions: retrieval quality and search efficiency.

*Search efficiency* is assessed primarily by the number of comparisons needed to retrieve a result for a given query. Unlike throughput, this metric is agnostic to implementation and hardware, offering a fair basis for comparing algorithmic efficiency.

- **Number of Comparisons**: Every time the $q$-Metric VP-Tree visits a node. Reflects the computational cost related to search speed.
- **Queries Per Second (QPS)**: Measures the throughput of the algorithm, indicating how many queries can be processed per second under the current configuration.

*Retrieval quality* is evaluated using metrics that capture not just whether relevant results are retrieved, but how well their ordering is preserved. In particular, we rely on Recall@k and Rank Order, which – unlike recall – penalizes deviations in the relative ranking of retrieved results. This is crucial in downstream tasks such as recommendation, where the order of results matters. The metrics used in our evaluation are summarized below:

- **RankOrder@k**: Measures how well the approximate method preserves the original ordering of the true nearest neighbors. Let $\mathcal{N}_k^{\text{true}}(y)$ be the true $k$-nearest neighbors of a query $y$, and $\mathcal{N}_k^{\text{approx}}(y) = \{x_1, \ldots, x_k\}$ the corresponding approximate result. Let $\pi(x_i, \mathcal{N}_k^{\text{true}}(y))$ denote the position of $x_i$ in the true result $\mathcal{N}_k^{\text{true}}(y)$ (or $k+1$ if not found). The metric is defined as:

$$\textbf{Absolute RankOrder@k} \tag{55}$$

$$(y) = \sum_{i=1}^{k} \left| i - \pi(x_i, \mathcal{N}_k^{\text{true}}(y)) \right| \cdot \frac{1}{k} \tag{56}$$

  Lower values indicate better rank preservation, with 0 being optimal.

  In addition, a variation of the rank order that takes into account the total number of points in the dataset is also used:

$$\textbf{Relative RankOrder@k} \tag{57}$$

$$(y) = \sum_{i=1}^{k} \left| i - \pi(x_i, \mathcal{N}_k^{\text{true}}(y)) \right| \cdot \frac{100}{nk} \tag{58}$$

  where $n$ is the size of the indexed points, i.e $|X|$. This measure expresses rank order but now as a percentage of the points available for retrieval.

- **Recall@k**: Measures the proportion of true $k$-nearest neighbors that are successfully retrieved by the approximate method. Let $\mathcal{N}_k^{\text{true}}(y)$ be the true $k$-nearest neighbors of a query $y$, and $\mathcal{N}_k^{\text{approx}}(y)$ the corresponding approximate result. The recall is defined as:

$$\texttt{Recall@k}(y) = \frac{|\mathcal{N}_k^{\text{true}}(y) \cap \mathcal{N}_k^{\text{approx}}(y)|}{k} \tag{59}$$

This metric ranges from 0 to 1, where 1 indicates that all true neighbors were retrieved. It reflects the *coverage* of the ground-truth neighbors in the approximate result.

### E.2 Canonical Projection $P_q^\star$

We validate the theoretical properties of searching in $q$-metric and ultrametric spaces presented in Section 2. In particular, we confirm our complexity claims (C1), the preservation of nearest neighbors, and the stability of rank order under projection (C2).

In order to do so, we use the Canonical Projection $P_q^\star$ presented in Section 3 to project distances imposing $q$-metric structure on FashionMNIST and GloVe. For these two datasets we use four different dissimilarities. After projecting dissimilarities, we search using the $q$-metric VP tree as described in Appendix B.

Figures 10a and 10b (first row), show that the number of nodes visited during search decreases monotonically with increasing $q$, reaching the theoretical minimum of $\log_2(n)$ at $q = \infty$. This directly confirms Theorem 1 and supports claim C1. Additionally, a rank order of zero across all queries–for a wide range of $q$ values (excluding $q = \infty$) and across all dissimilarities– confirms claim and C2, as established in Lemma 2 and Proposition 1. At the same time, this nearest neighbor preservation is also observed at recall, showing perfect matches at $k = 1$ for moderate $q$ values.

At $q = \infty$, the projection may introduce spurious optima not included in the original nearest neighbor set. Although the original nearest neighbours are still a solution in the transformed space, in practice we observe that the spurious optima at $q = \infty$ indeed affect accuracy. A similar effect occurs for large $q$, as distances between points can become artificially close, imitating this behavior.

Although our theoretical guarantees focus on the $k = 1$ nearest neighbor case, an empirical preservation of locality observed in Figures 10a and 10b. We evaluate this by searching for $k = 5$ and $k = 10$ neighbors using the projected distances, as shown in the second and third rows of Figures 10a and 10b. While the number of comparisons does not decrease as rapidly as in the $k = 1$ case, the method consistently yields improvements across all values of $q$.

### E.3 Approximating the canonical projection with $\Phi(x; \theta^\star)$

We analyze how well the learned distances $\hat{E}_q(x, x') = \|\Phi(x; \theta^\star) - \Phi(x'; \theta^\star)\|$, described in Section 4, reproduce the properties of the true q-metric distances $E_q(x, x')$.

As described in Section 4, the learning process minimizes two loss terms: the stress $\ell_D$, which measures the squared error between the learned distances and the true projected distances, and the triangle inequality violation $\ell_T$, which penalizes failure to satisfy the $q$-triangle inequality.

Figure 11 shows that $\ell_D$ increases monotonically with $q$, consistent with the trend observed in Section E.4 where retrieval accuracy declined at high $q$ values. This indicates that approximating the Canonical Projection becomes more difficult as $q$ increases.

We observe the violation of the $q$-triangle inequality $\ell_T$ decreases with increasing $q$. However, the corresponding increase in accuracy metrics like rank order shows an opposite behavior is observed. That is, we observe $\ell_D$ to be more correlated with downstream performance than the satisfaction of the $q$-triangle inequality as measured by $\ell_T$.

Predicted and ground truth distances in Figure 12 show similar distributions for training and testing distances albeit for a generalization gap. Again as $q$ increases, so does the approximation error.

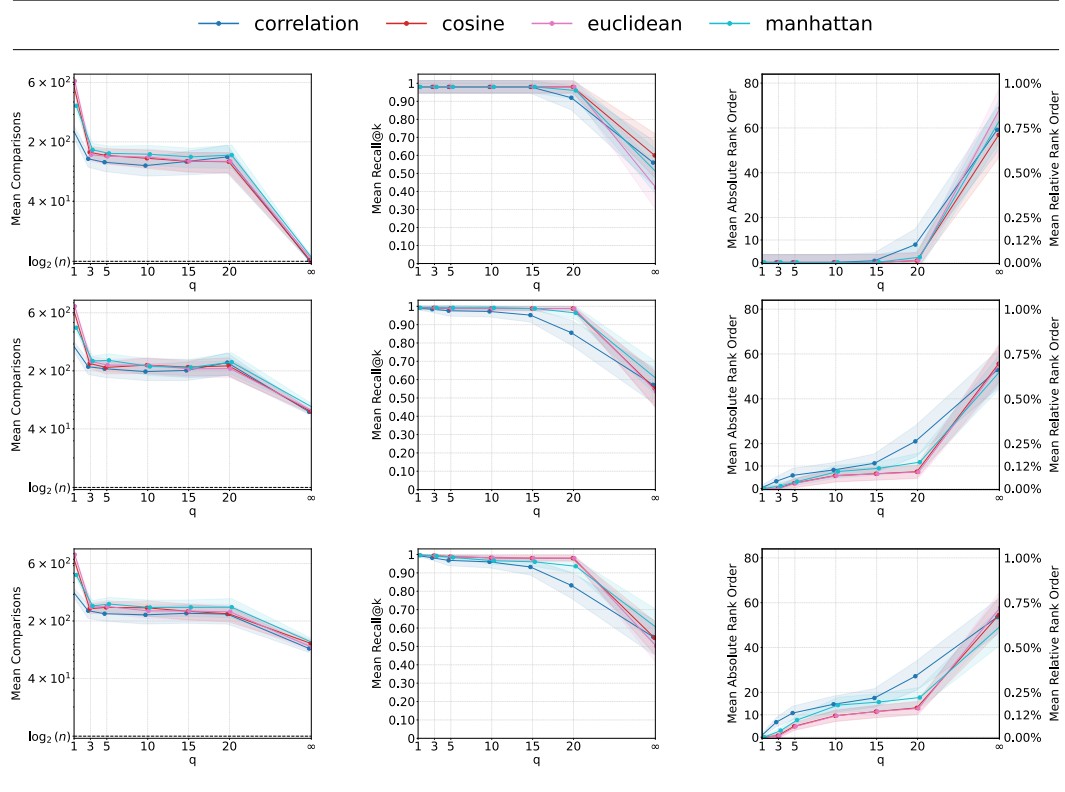

(a) $n = 1{,}000$ points of Fashion-MNIST

Figure 10: Number of comparisons and rank order across different dissimilarities when searching after applying Canonical Projection when a query point is added ($E_q$). The search was performed with a $q$-VPTree. Solid lines denote the mean and shading the standard deviation computed across queries. Each row shows results for $k$-nearest neighbors, with $k = 1, 5, 10$ from top to bottom.

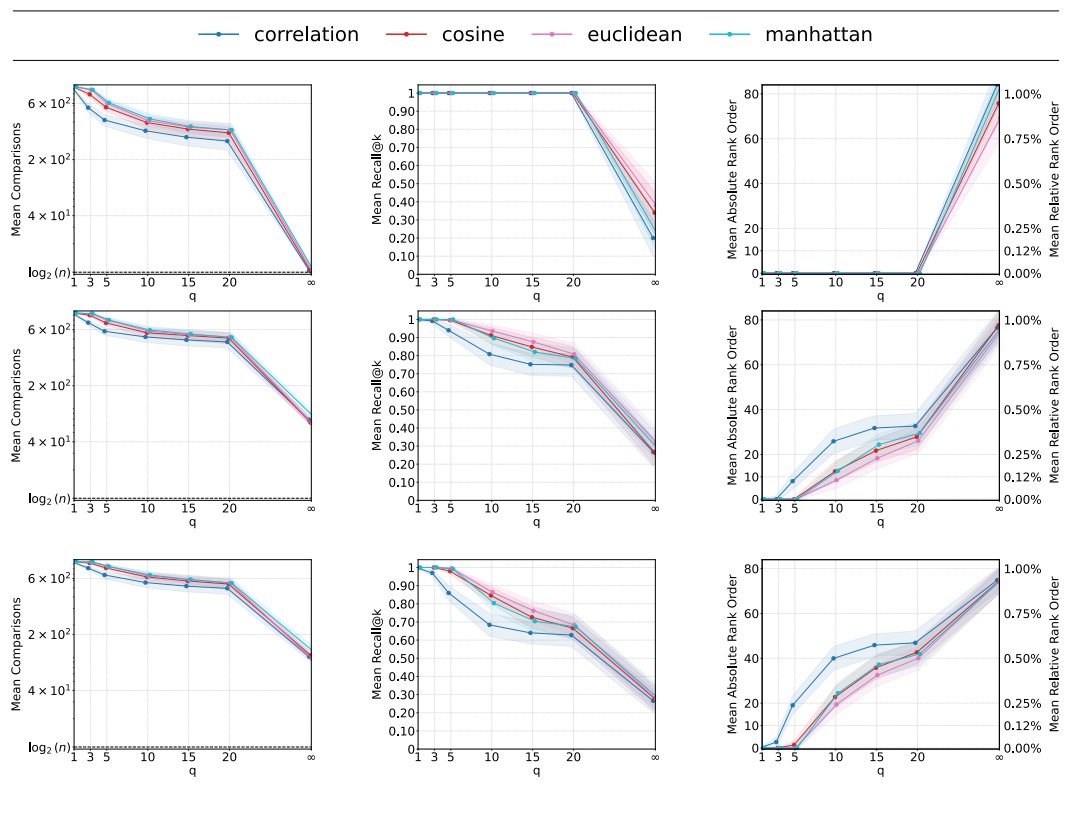

(b) $n = 1,000$ points of GloVe

Figure 10: Number of comparisons and rank order across different dissimilarities when searching after applying Canonical Projection when a query point is added ($E_q$). The search was performed with a $q$-VPTree. Solid lines denote the mean and shading the standard deviation computed across queries. The $k$-nearest neighbors are listed from top to bottom for $k = 1, 5, 10$

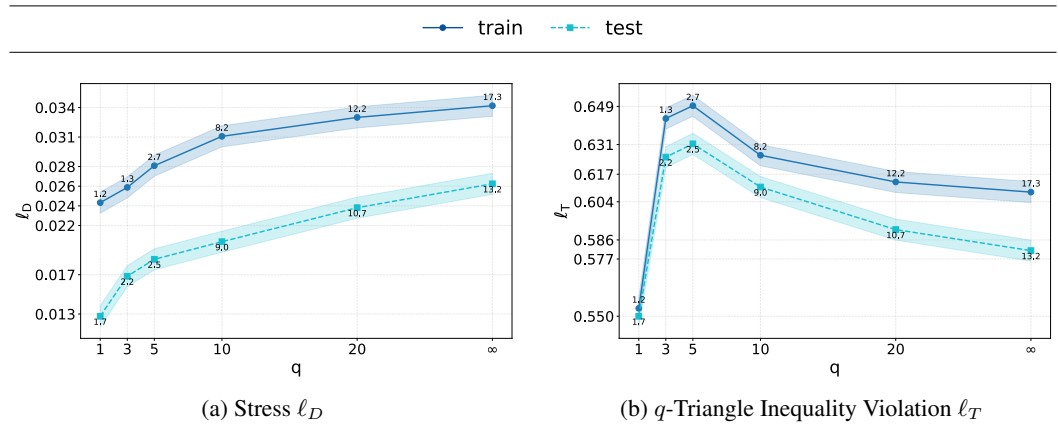

(a) Stress $\ell_D$            (b) $q$-Triangle Inequality Violation $\ell_T$

Figure 11: Values of the Stress $\ell_D$ and the $q$-triangle inequality regularizer after the training process. Labels of each point represent average Rank Order. This case corresponds to the learning process performed over the Fashion-MNIST dataset for euclidean distance.

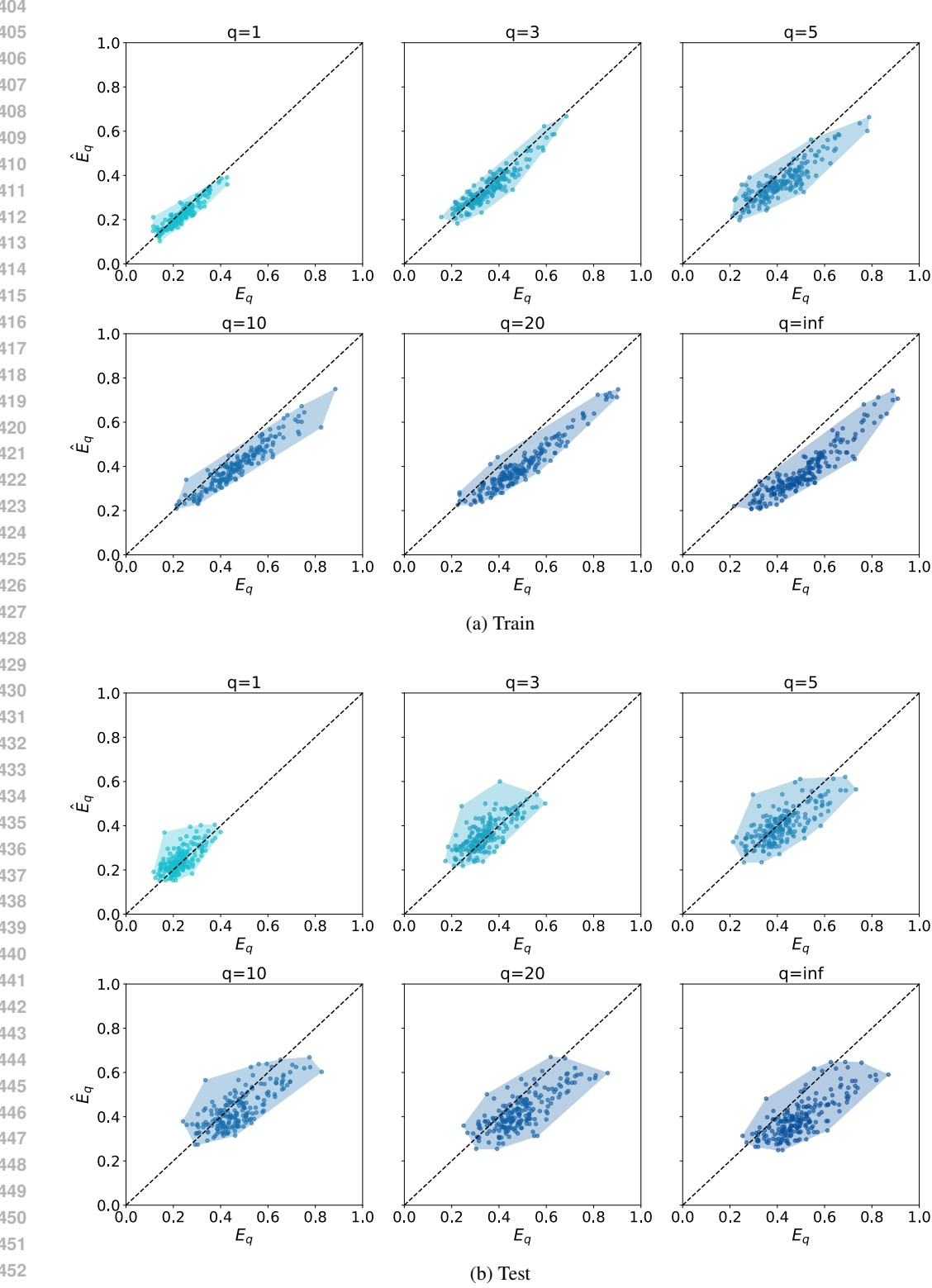

Figure 12: Distribution of the distance to Nearest Neighbor. In the X-axis $E_q$ depicts distance values obtained after projecting when a query point is added. The Y-axis shows the learned approximation of the Canonical projection $\hat{E}_q$. The dashed line represents the perfect match between projected and approximated $x = y$.

## E.4 Searching with $\phi_q(\cdot, \theta^\star)$

In this section we demonstrate that the speedup observed in the exact projection experiments of Section E.2 is also attained when replacing the Canonical projection with the learned map $\phi_q(\cdot, \theta^\star)$, at the cost of small errors in search results.

Figures 13a and 13b show a consistent reduction in the number of comparisons as $q$ increases, mirroring the trend observed in the exact projection experiments. This reduction is accompanied by a moderate increase in rank error and a decrease in recall, suggesting that the retrieved neighbors remain close but are not always exact. Nevertheless, cases with recall above 0.9 still yield substantial speedups. Since recall captures only exact matches and the method guarantees order preservation only for the 1-nearest neighbor, rank order can provide a complementary view of performance.

Despite the degradation of the approximation quality for high $q$ values, the retrieved neighbors remain within the top 80 nearest, which corresponds to less than $1\%$ of the indexed dataset. When higher precision is required, setting $q = 10$ can yield a two-orders-of-magnitude speedup (Figure 13a) while returning neighbors ranked around 10th, corresponding to a relative error close to $0.12\%$. These results demonstrate that the learned embeddings preserve local structure to a satisfactory extent.

The results also extend to k-nearest neighbor search with $k > 1$. While the reduction in comparisons is less pronounced than for $k = 1$, the method maintains a consistent speedup across values of $q$. Rank Order remains low, and in some cases improves as $k$ increases, suggesting that the learned map preserves small-scale neighborhood structure reasonably well even on these datasets.

Examples of retrieval results were also generated. For each dataset, queries were selected from varied categories. In the Fashion-MNIST case (Figure 14), each panel shows the original query image, its true nearest neighbor, and the result returned by Infinity Search. While the exact nearest neighbor was not retrieved in some cases—such as those involving sandals or sneakers—the returned items consistently belonged to the same category as the query. For GloVe-200 text embeddings, Figure 14b includes several exact matches, as well as examples where the retrieved word preserved the semantic meaning of the query.

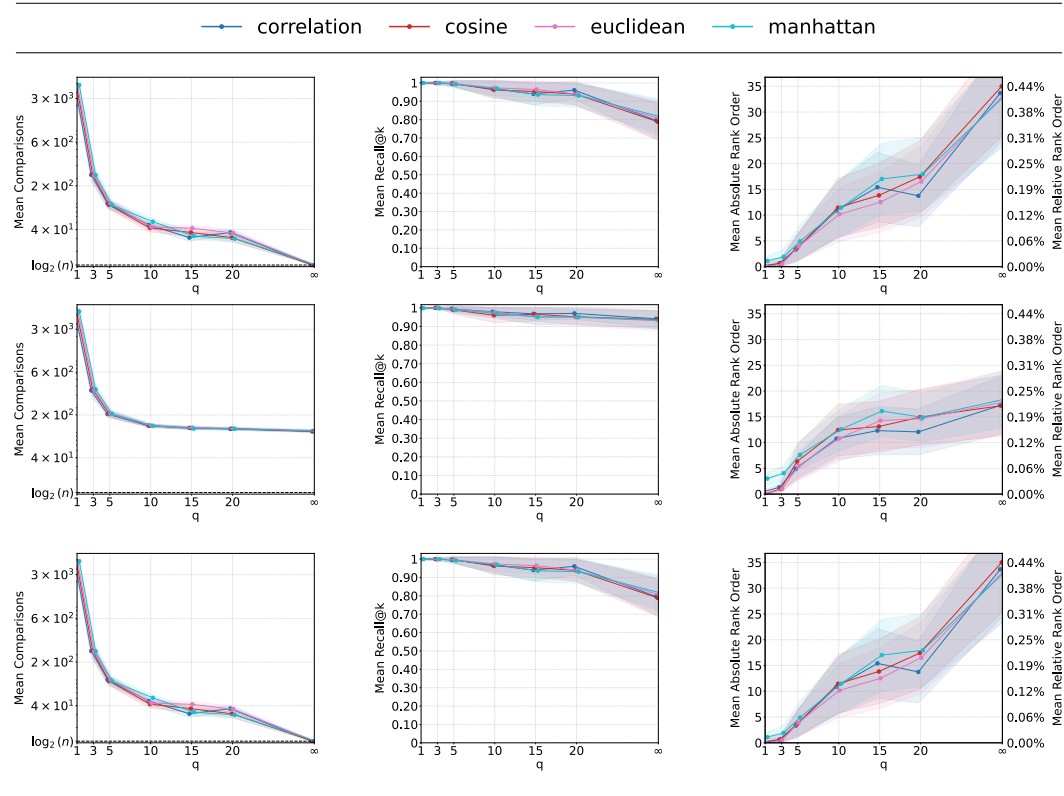

(a) $n = 10{,}000$ points of Fashion-MNIST

Figure 13: Number of comparisons and rank order when searching after approximating the Canonical Projection $\hat{E}_q$, with the learned map $\Phi(x; \theta)$. Solid lines denote the mean and shading the standard deviation computed across queries. The $k$-nearest neighbors are listed from top to bottom for $k = 1, 5, 10$

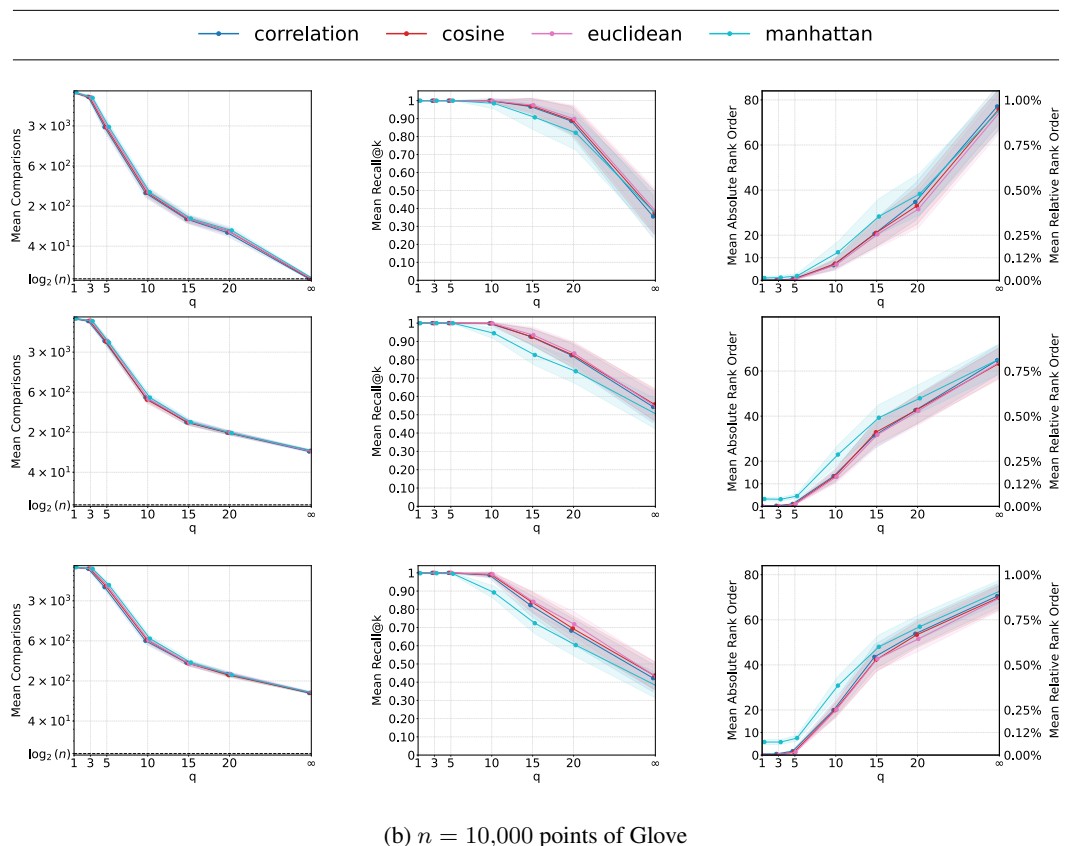

(b) $n = 10{,}000$ points of Glove

Figure 13: Number of comparisons and rank order when searching after approximating the Canonical Projection $\hat{E}_q$, with the learned map $\Phi(x; \theta)$. Solid lines denote the mean and shading the standard deviation computed across queries. The $k$-nearest neighbors are listed from top to bottom for $k = 1, 5, 10$

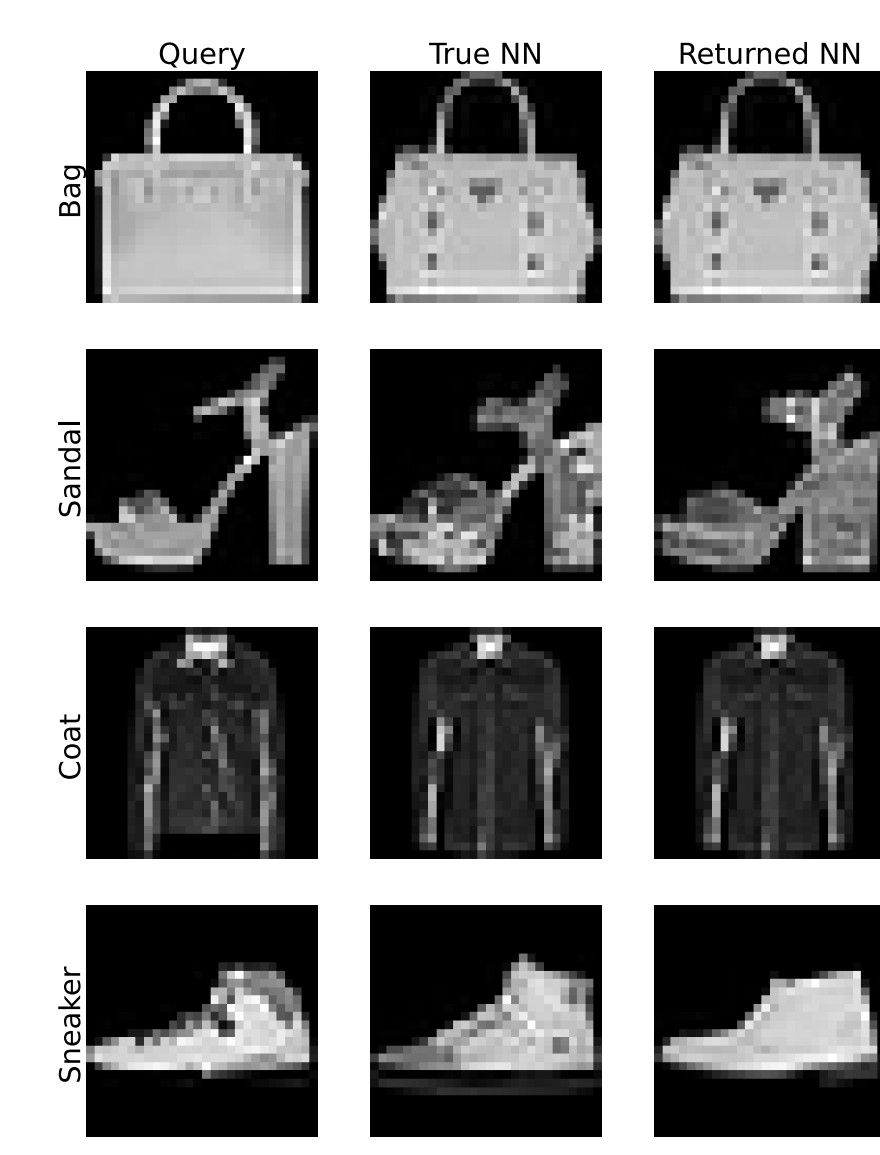

(a) Fashion-MNIST

| Category | Query | True NN | Returned NN |
|----------|-------|---------|-------------|
| **Animals** | dog | dogs | dogs |
|  | lion | wolf | wolf |
| **Colors** | blue | pink | purple |
|  | red | pink | purple |
| **Clothing** | pants | jeans | jeans |
|  | shirt | shirts | worn |
| **Tools** | drill | drilling | drilling |
|  | hammer | throw | flame |

(b) GloVe

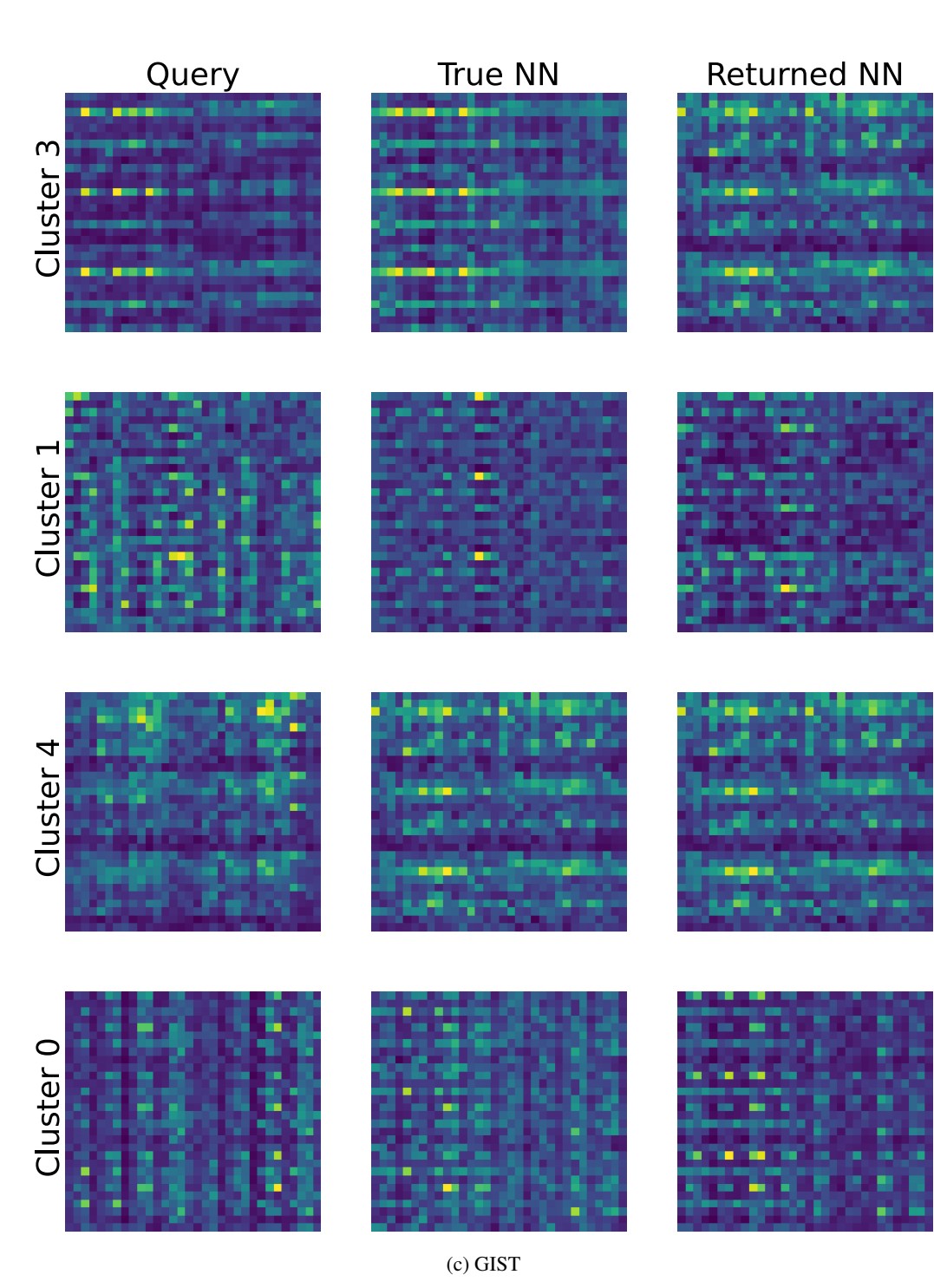

(c) GIST

Figure 14: Retrieval of dataset items with Infinity Search ($q = 5$).

Although Proposition 8 ensures preservation of the nearest neighbor structure, it also notes that the projected nearest neighbor may not remain unique. This ambiguity can lead to mismatches during retrieval. The effect appears in theoretical settings, as seen in Figures 10a and 10b, where rank order the rank order exhibits a sharp increase at $q = \infty$. A similar trend can be observed in the approximate setting, shown in Figures 13a and 13b. In Figure 15, the distribution of projected $q$-metrics shows a clustering effect as $q$ increases. Distances become more concentrated around mean, while the frequency of close values also increases. This suggests that distances between points become closer, making true nearest neighbors more difficult to discern.

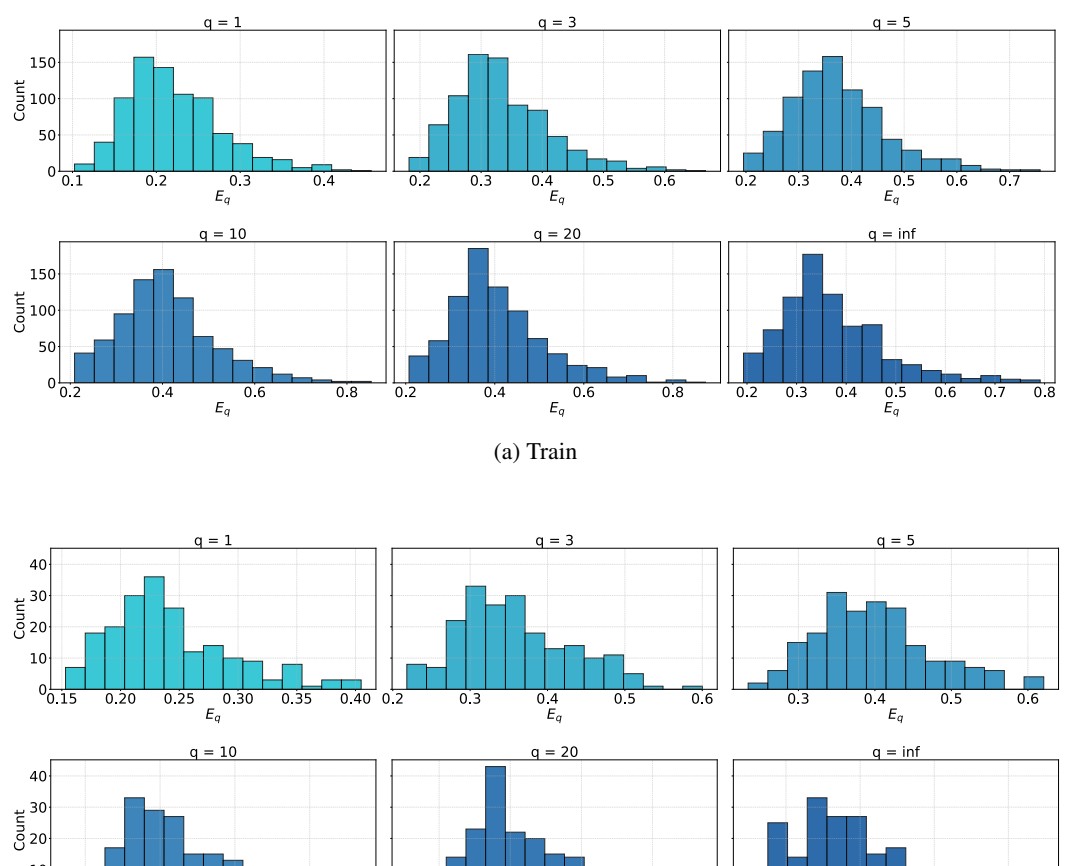

(a) Train

(b) Test

Figure 15: Histogram of the distance to Nearest Neighbor. In the X-axis $E_q$ depicts distance values obtained after projecting when a query point is added. The Y-axis shows number of counts for that distance bin.

To address this, one can extend the nearest neighbor set using the Canonical Projection and then prune it to avoid loss in accuracy. This motivates a two-stage modification of the Infinity Search algorithm:

- **Broad Search**: The Infinity Search algorithm is used to retrieve an initial candidate set of $K$ nearest neighbors. This will retrieve close neighbors in the $q$-metric space.

- **Specific Search**: Once the list of $K$ candidate neighbors is available, the original distance $D$ is used to retrieve the $k$ real nearest neighbors.

This two-stage retrieval strategy is used in some ANN methods, including HNSW Malkov and Yashunin (2020). As shown in the theoretical and approximate experiments of Sections E.2 and E.4, the Canonical Projection preserves locality but not the exact order of nearest neighbors. This makes the two-stage approach suitable for improving accuracy.

Figures 16a and 16b confirm the improvement, showing higher recall and more accurate rank alignment compared to earlier approximations. The gains are particularly notable in rank order, with a 3 to 4 times reduction in error. As expected, this comes with a decrease in speedup, since the method processes a larger candidate set and computes original distances during Specific Search. Although the logarithmic comparison bound ($\log_2(n)$) no longer holds, the resulting speed remains competitive. The original Infinity Search can be recovered by setting $K = k$, while choosing $K > k$ offers additional flexibility to trade off speed and accuracy depending on the requirements of the searching problem.

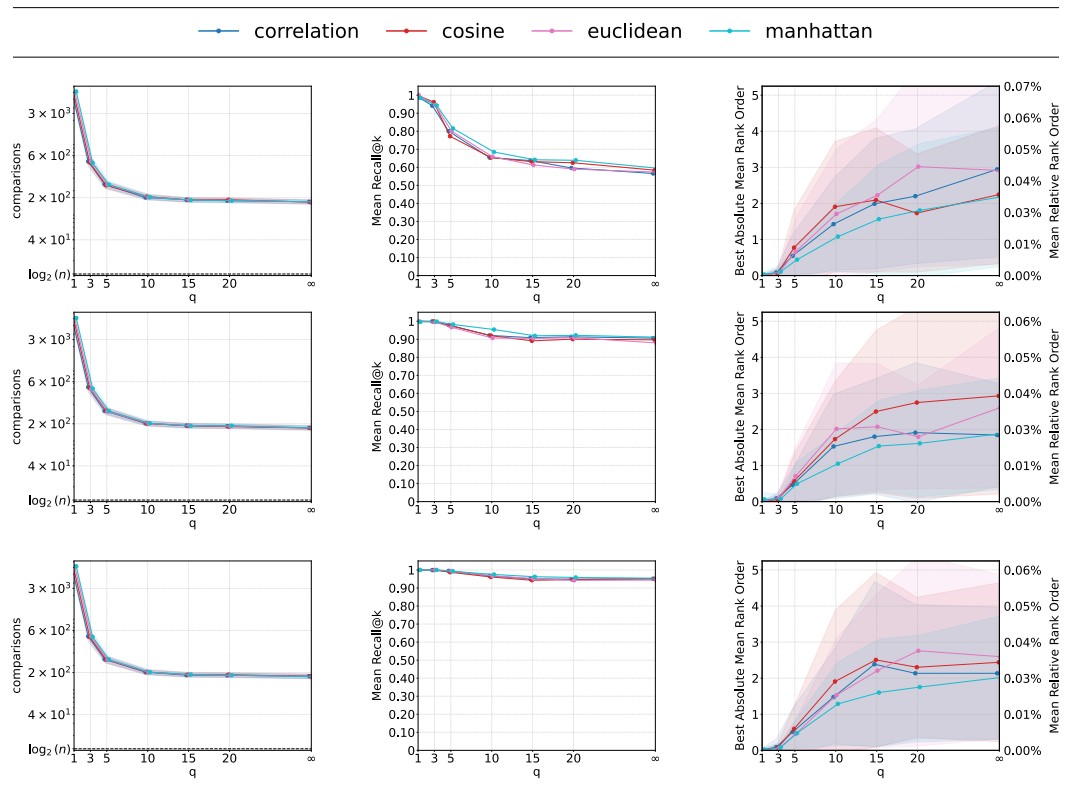

(a) $n = 10,000$ points of Fashion-MNIST

Figure 16: Number of comparisons, Recall@k and Rank Order when searching with a two-stage retrieval Infinity Search. Solid lines denote the mean and shading the standard deviation computed across queries. The $k$-nearest neighbors are listed from top to bottom for $k = 1, 5, 10$

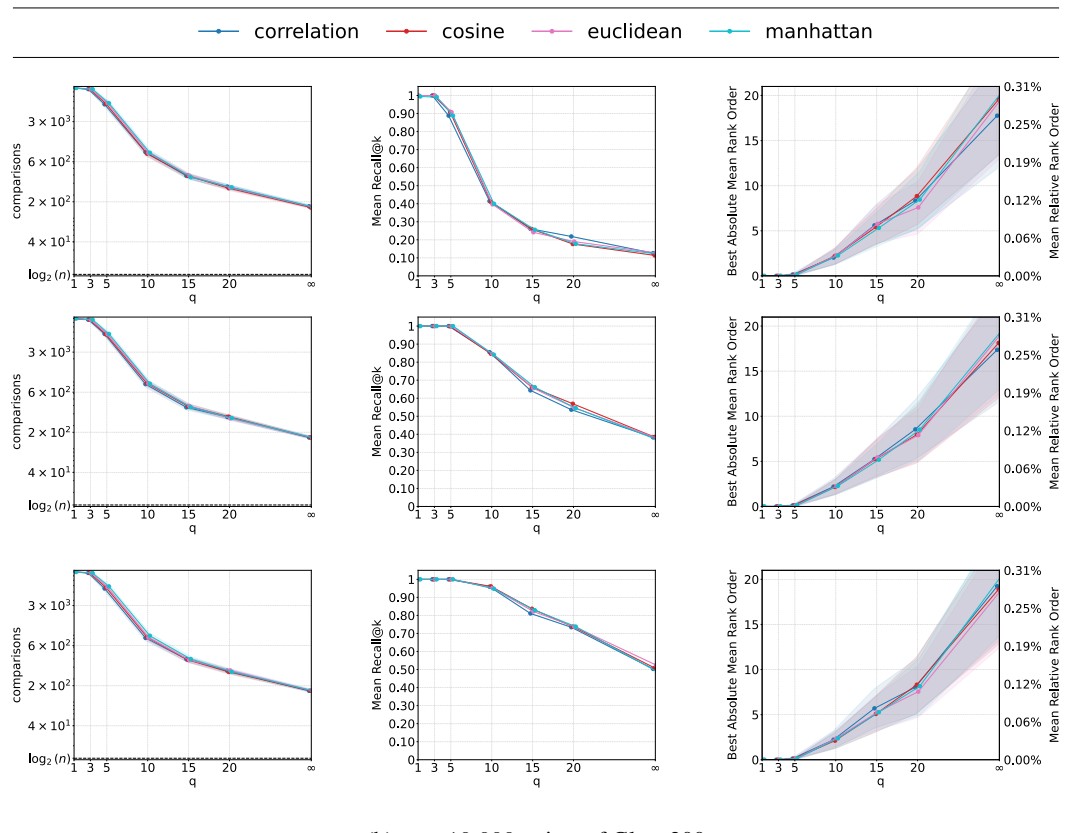

(b) $n = 10,000$ points of Glove200

Figure 16: Number of comparisons, Recall@k and Rank Order when searching with a two-stage retrieval Infinity Search. Solid lines denote the mean and shading the standard deviation computed across queries. The $k$-nearest neighbors are listed from top to bottom for $k = 1, 5, 10$

### E.6  INFINITY SEARCH SCALES

In this section we examine how the method scales with dataset size and dimensionality. The current implementation is not engineered for industrial deployment, but we seek to characterize behavior as problem size increases. We index subsamples $n \in \{10K, 50K, 100K, 500K, 1M, 5M\}$ from Deep1B Babenko and Lempitsky (2016), holding the rest of the pipeline fixed to isolate scaling effects.

The projection $P_q^*$ is trained once on a fixed set of 100K points. After training, each target subset is projected prior to indexing; no additional tuning is performed. This mimics an inductive setting in which a single model serves increasingly large corpora with constant per-point projection cost.

As shown in Figure 17, the search stage exhibits competitive scalability, following a sub-logarithmic trend in $n$. In terms of accuracy, *Rank Order* more clearly captures how error grows with index size. We observe degradation when applying a model trained on 100K points to 1M–5M points, consistent with inference mismatch from training on a limited subset. Despite this shift, the embeddings remain meaningful on large validation sets, indicating good inductive transfer without retraining.

Regarding construction cost, Figure 17 shows an essentially linear build time, $O(n)$, which is expected for tree-based indexing. Overall, these results indicate: (i) favorable search scaling, (ii) predictable accuracy drift with growing $n$ under fixed training size, and (iii) linear build complexity. Extending training to larger or stratified subsets, or enabling lightweight incremental updates, is a natural direction for future work.

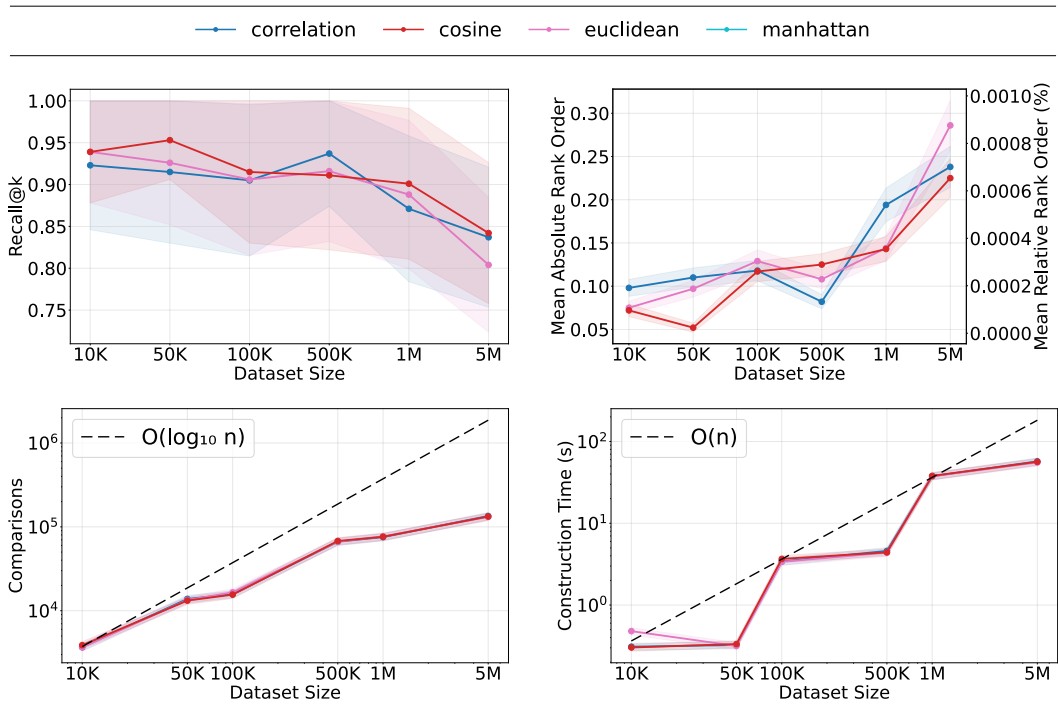

Figure 17: Infinity Search results on searching $n \in \{10K, 50K, 100K, 500K, 1M, 5M\}$ points of Deep1B-96 with Euclidean distance.

The recent increase in the expressiveness of embeddings has made performance on high-dimensional data a key requirement. Moreover, traditional VP-trees have struggled to prune effectively in high dimensions. We hypothesize that this is due to concentration of measure, a consequence of the *curse of dimensionality*, which causes pairwise distances to concentrate and weakens pruning bounds. Figure 18 reports Infinity Search results on GloVe Pennington et al. (2014) with increasing dimensionality, $\{50, 100, 200, 300\}$. The number of comparisons grows monotonically with dimensionality, supporting the hypothesis that pruning is easier in lower dimensions than in higher-dimensional

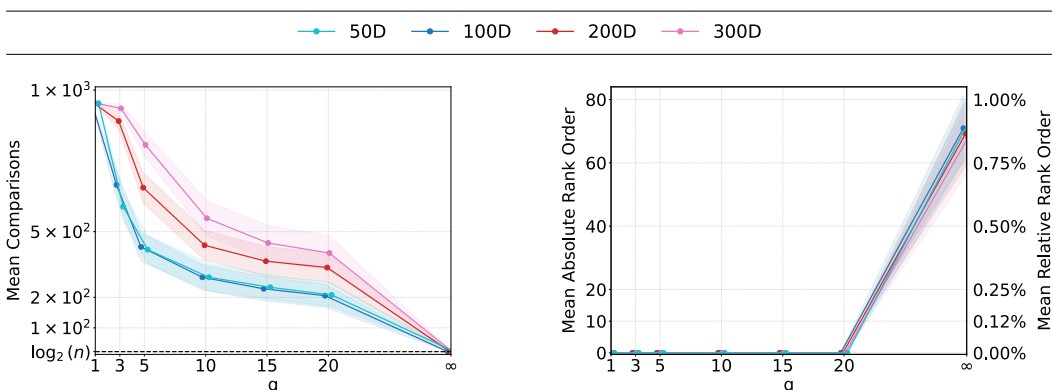

Figure 18: Number of comparisons and rank order across different dimensions when searching after applying Canonical Projection when a query point is added ($E_q$). Solid lines denote the mean and shading the standard deviation computed across queries. The dataset used was GloVe with $n = 1,000$ points and the dissimilarity the standard euclidean distance.

setups. As in our scaling experiments, $P_q^*$ is trained once per dimension and then applied inductively with a fixed per-point projection cost, isolating the effect of dimensionality.

However, our method attains logarithmic search complexity with respect to $n$ regardless of dimensionality. In addition, Rank Order remains consistent with the theoretical analysis in Section E. Taken together, these results indicate that both search complexity and accuracy are preserved even as dimensionality increases.

### E.7 INFINITY SEARCH COMPETES IN ANN-BENCHMARKS

Infinity Search offers a configurable trade-off between query throughput and recall. To assess its competitiveness against state-of-the-art ANN methods, we evaluated it within the ANN-Benchmarks framework Aumüller et al. (2018). We ran experiments on five datasets provided by the library and compared against a wide set of algorithms chosen for their balance of speed, accuracy, and open-source availability.

In both theoretical analysis and empirical benchmarks, Infinity Search consistently accelerates nearest-neighbor queries across all tested dissimilarities. On moderate-dimensional datasets such as Fashion-MNIST and GIST (Figure 19), it delivers a clear speedup by, in some cases, sacrificing perfect accuracy. Remarkably, on the high-dimensional Kosarak dataset—with Jaccard dissimilarity—Infinity Search outperforms competing methods by an even wider margin. This supports the flexibility of the method when less popular dissimilarities are required.

Across all datasets, it offers a favorable speed–accuracy trade-off for the $k = 1$ nearest-neighbor task. For larger neighborhood sizes ($k \in \{5, 10\}$), the increased comparison overhead prevents it from always leading in Recall@k; nevertheless, its performance remains competitive. Note that, since $n = 10,000$ in these experiments, modest rank-order errors at extreme speeds still correspond to few misplaced neighbors.

Overall, Infinity Search is a viable alternative when fast retrieval is required or non-Euclidean or less structured similarity are used.

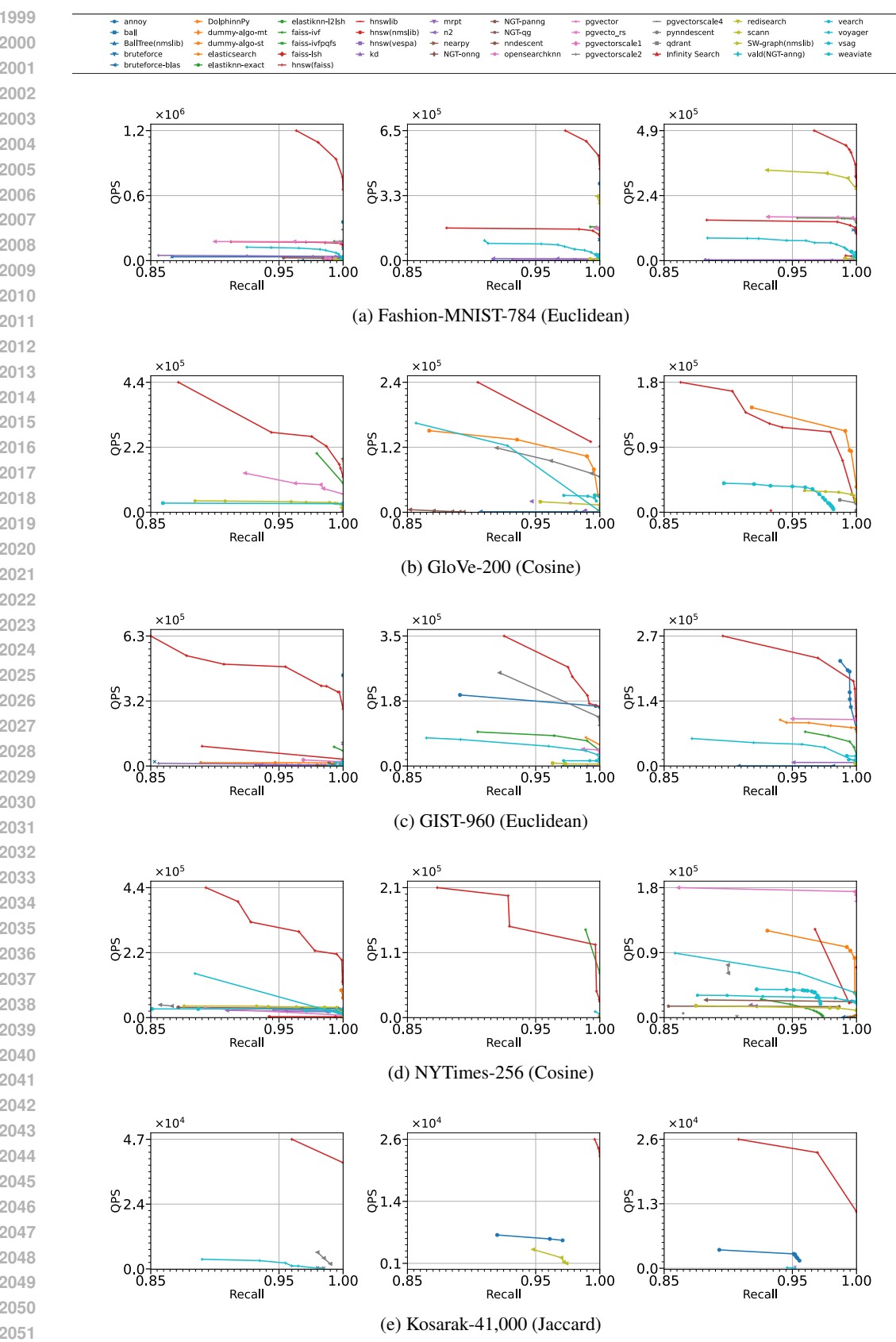

Figure 19: Recall@k vs queries-per-second (QPS) across datasets. Columns within each row are $k = 1, 5, 10$

