# OpenReview forum: "Infinity Search: Approximate Vector Search with Projections on q-Metric Spaces"
_ICLR.cc/2026/Conference — ICLR 2026 Conference Desk Rejected Submission_

### Official Review · Reviewer_MHeW · 2025-11-03

**Soundness:** 1
**Presentation:** 1
**Contribution:** 1
**Rating:** 2
**Confidence:** 3

**Summary:**

Authors propose to learn a function to map vector embeddings into separate embeddings such that their Euclidean distances estimate the distances of the q-metric projections of the original embedding. The advantage of doing so is that a q-metric can approximate an ultrametric and with the ultrametric k-NN search is solvable in logarithmic time using a VP-tree. Authors also claim to support transformation for arbitrary similarities, but the provided code supports only L_2 (with possible extensions to p-norms).

The idea seems to be very appealing. However, vector spaces do not seem to have non-trivial ultrametric distances. Thus, intuitively, the closer we approximate an ultrametric, the more degenerate/trivial such approximation should become.

As much as I like the idea, I find parts of the paper impenetrable. For example, in L103 authors start talking about graphs without properly defining them. Moreover, graphs are the central part of the section 3, which I could not understand.

However, I read the code, and it is relatively straightforward: training a projection neural network, embed data, and search it using a VP-tree. In that, the code has multiple serious issues (see the questions) and I am rather confident that it cannot work properly.  I hope authors can clarify this by answering some of my questions.

Moreover, despite author’s claims of integrating with ANN-benchmark, they have a standalone pipeline which raises questions about the accuracy of the evaluation (in fact it does not compute recall or something like rank mean difference).

**Strengths:**

1. An interesting idea.
2. Strong **reported** results.
3. Code is provided.

**Weaknesses:**

Please see the summary. Basically, I do not think the code is doing what it is supposed to do and the evaluation is likely incorrect.

**Detailed comments:**

L031 Hanov does not seem to be an appropriate reference for this statement. These are more appropriate classic references:

1. Kevin S. Beyer, Jonathan Goldstein, Raghu Ramakrishnan, and Uri Shaft. When is ”nearest neighbor” meaningful? 1999.

Same mistake in L738: the classic VP-tree references:

   * Jeffrey K. Uhlmann. 1999. Satisfying general proximity/similarity queries with metric trees
    * Stephen M Omohundro. Five balltree construction algorithms, 1989. ICSI Technical Report TR-89-063,

2. Roger Weber, Hans-Jorg Schek, and Stephen Blott. A quantitative analysis and performance study for similarity-search methods in high-dimensional spaces.  VLDB’98,

L053 However, the data is what the data is and most problems in vector search involve dissimilarity functions -> This is an old reference to back up a claim about most problems.

L059-060
> In this context, it follows that the q-norm of a path is the q-root of the sum of the distances in each hop elevated to the power of q

It is completely unclear. The paragraph needs to be rewritten for clarity.

Section 3.1 and the part of section 3 before 3.1 are not understandable, in particular, because they talk about graphs that aren’t defined.

**Questions:**

**L194-196**
Do you have examples of non-trivial ultrametric vector spaces? I think these don’t exist. Thus, Theorem 1 does not apply to real world cases.

**L103**
> defines a fully connected weighted graph G= (X,D)
** L214-215 **
> We use the projection operator Pq to process the dataset G = (X,D) to produce a graph Gq = (X,Dq) such that any three points x,y,z ∈X satisfy the q-triangle inequality.

What are exactly these graphs?

**L488** Why is your evaluation code is not interoperable with ANN-benchmarks? It is clearly a standalone evaluation toolkit.

1. Why is your evaluation code is not interoperable with ANN-benchmarks? It is clearly a standalone evaluation toolkit.

2. Does your own code really compute recall or **relative position error**?

**Why does loss in section 4** not match the actual loss in the code?

There are multiple issues with the code. In each epoch you take the whole dataset for one loss and a batch for another, which is very non-standard, and I doubt it works.
More importantly, the second loss computation can't be correct in my opinion for several (not just a single reason).

See comments below:
```
 D0 = torch.cdist(X, X, p=2)
 M = self.fermat_gpu_exact(D0, q)
 M = (M - M.min())/(M.max()-M.min())
# This computers an all-to-all matrix of distances not pairwise distances!
def emb_dist(a, b=None): return torch.cdist(a, a if b is None else b, p=2)

# All data points are embedded
emb = model(X)
D_emb = emb_dist(emb)
mask = M.float()
# This loss might be reasonable though I don't quite understand how M is computed
loss_s = torch.sqrt(((D_emb - M)**2 * mask).sum() / mask.sum())
idx = torch.randint(0, n, (batch_size, 3), device=device)
i,j,k = idx.t()
# Problem 1: emb_dist does NOT compute pairwise distances.
# Problem 2: There is no exponentiation to the power of q here.
# Problem 3: If you computed pairwise distances correctly, due to the triangle inequality,
# d(i,j) + d(j, k) is always >= dist(i,k)
# You pass it through RELU and get something (mostly positive)
# Thus you will PENALIZE your embeddings for actually satisfying the triangle inequality
raw = emb_dist(emb[i], emb[j]) + emb_dist(emb[j], emb[k]) - emb_dist(emb[i], emb[k])
loss_t = F.relu(raw).min()
```

Shouldn’t run_batch_query accept embeddings as the first argument?
```

def run_batch_query(self):

return self.index.search_batch(1, self.kk, self.queries_np, self.queries_np, False)

```
In fact, this function eventually call search_batch, which reads embedding vectors from the same location as original vectors, but uses different offsets (since embedding dimensions is different):
```

inline std::vector<std::vector<int>>

VpTree::search_batch(int k,int topk,const float* qE,const float* qR,std::size_t N,bool retDist) const {
  std::vector<std::vector<int>> all(N);
  int dE=embed_.dim, dR=real_.dim;

  for(size_t i=0;i<N;++i) {
    auto r = search_core(k, qE+i*dE, qR+i*dR, retDist);
    if ((int)r.ids.size() > topk)
      r.ids.resize(topk);
    all[i] = std::move(r.ids);
  }
  return all;
}
```

---

> ### Author Response · Authors · 2025-11-30
>
> Dear Reviewer MHeW,
>
> We appreciate the reviewer’s effort in revising our paper. Several points appear to stem from misunderstandings of our implementation and the ANN-Benchmarks interface rather than substantive errors, and they do not invalidate our experimental results. Moreover, the code we provided is a minimal, self-contained script intended to illustrate how the algorithm operates on a small experiment; it is not the code used for the full set of experiments reported. If helpful, we can share the full ANN-Benchmarks integration code as well, although running the complete suite can be computationally intensive. We address each concern below.
>
> **C1: L194-196 Do you have examples of non-trivial ultrametric vector spaces? I think these don’t exist. Thus, Theorem 1 does not apply to real world cases.**
>
> Indeed, real datasets are not ultrametric, which motivates the ultrametric Projection (see L203-205) introduced in Section 3. As stated in the submission, this enables projecting any dataset and (dis)-similarity into a (non-trivial) ultrametric space.
>
> -L203-205: "Alas, dissimilarity metrics of interest do not satisfy the strong triangle inequality.
> Some are not even metric. Due to this mismatch, we propose here an approach to approximate nearest
> neighbor search based on the development of projection..."
>
> **C2: L103: What are exactly these graphs?**
>
> As explained in line 102, the graph is made of all points and has pairwise distances as edge weights.
>
> -L102: "The set X along with the set D containing all dissimilarities d(x, y) for all points x, y ∈ X
> defines a fully connected weighted graph G = (X, D)"
>
> **C3: L488: Why is your evaluation code not interoperable with ANN-benchmarks? It is clearly a standalone evaluation toolkit.**
>
> The submitted code is in fact interoperable with ANN-Benchmarks. We did use the ANN-Benchmarks evaluation in the reported results. The code included in the supplementary material is a minimal self-contained reproduction of experiments of Figures 3 and 4, which do not involve the full ANN-Benchmarks suite.
>
> **C4: Does your own code really compute recall or relative position error?**
>
> The code computes relative error; we omitted other metrics for brevity.
>
> **Code comments:**
>
> **C5: Problem 1: This computes an all-to-all matrix of distances not pairwise distances!**
>
> This is not correct; Torch cdist computes pairwise distances (see [official documentation](https://docs.pytorch.org/docs/stable/generated/torch.cdist.html)).
>
> **C6: All data points are embedded**
>
> We do not understand the concern of the reviewer; can you please clarify? Notice that X contains only the indexed set of points passed to the fit() method [Code L183]. Since the provided code is intended for small datasets, we omit batching.
>
> **C7: This loss might be reasonable though I don't quite understand how M is computed**
>
> M is the result of the projection in Section 3, which obtains q-metric structure. The loss, as described in equation (10) at line 305 of the main body, enforces pairwise Euclidean distances to resemble pairwise projected distances. The “mask” term is a scalar and does not affect  training.
>
> **C8: Problem 2: There is no exponentiation to the power of q here.**
>
> The exponentiation of q does not happen in the initial distances; it happens during the projection described in the main body [Code L47] “M = D.pow(q)”. Notice that the q-norm does not satisfy the triangle inequality, as specified in page 3 footnote.
>
> **C9: Problem 3: If you computed pairwise distances correctly, due to the triangle inequality,**
>
> Notice that the multiplier for this line is 0, and therefore is never used: [Code L111] “lambda_triangle = 0”. This is a remnant of a previous version of the code and it is not used in the ANN experiments, and it will be removed.
>
> **C10: Problem 4: Shouldn’t run_batch_query accept embeddings as the first argument?**
>
> Not necessarily; this is not required in ANN-benchmarks library guidelines and available implementations. Embeddings are passed to the prepare_query method, which is common practice (see e.g. Kgn):
>
> def prepare_query(self, q, n):
>     if self.metric == 'IP':
>         q = q / np.linalg.norm(q)
>     self.q = q
>     self.n = n
>
> def run_prepared_query(self):
>     self.res = self.searcher.search(self.q, self.n)
>
> **C11: Problem 5: which reads embedding vectors from the same location as original vectors, but uses different offsets (since embedding dimensions is different):**
>
> As specified in Appendix E.5, search_batch uses original vectors for a final reranking; this is a common practice in ANN methods (see e.g. FAISS IVF+PQ with refinement in the [FAISS wiki: Indexing 1M vectors](https://github.com/facebookresearch/faiss/wiki/Indexing-1M-vectors)).

---

### Official Review · Reviewer_Hq62 · 2025-11-03

**Soundness:** 2
**Presentation:** 2
**Contribution:** 1
**Rating:** 0
**Confidence:** 5

**Summary:**

This paper focuses on the problem of nearest neighbor search (NNS) and proposes a solution based on metric embedding. In this solution, data samples are first projected into a q-metric space. A projection is then learned from these samples, and the dataset is transformed into the learned q-metric space to build a VP-tree. Finally, NNS can be performed using this VP-tree.

**Strengths:**

+ The paper studies a meaningful problem.
+ Experiments are conducted on benchmark datasets.
+ The proposed solution is supported by some theoretical analysis.

**Weaknesses:**

+ The motivation is unconvincing.
+ The technical novelty is limited.
+ The proposed solution has several (potential) limitations.
+ The evaluations are incomplete.
+ The related work is not comprehensive enough and should not be deferred to the appendix.

**Questions:**

Q1. The motivation for the proposed method is unclear. Given that existing solutions like HNSW already achieve $O(logn)$ time complexity for Approximate NNS (ANNS), the necessity of a new approach is not justified. This is especially true when the proposed solution does not improve upon this complexity and introduces several potential limitations.

Q2. The technical novelty of the proposed method is limited for several reasons.
+ The theoretical foundation heavily overlaps with prior work. The main theorems (e.g., Theorem 2) and the key idea of using a q-metric VP-tree are very similar to those established in the 2015 paper "Metric Representations of Network Data."
+  Beyond this closely related work, metric embedding has been commonly used in ANNS. There are other options, such as the Hierarchically Well-Separated Tree (HST). It is unclear why such solutions are not considered or why the proposed method is superior.
+ It offers no complexity improvement. The method does not advance the state-of-the-art in worst-case time complexity, which remains $O(logn)$.

Q3. The proposed solution is designed for a static environment. The approach does not account for dynamic data updates, a critical requirement in real-world scenarios. Besides, if the similarity function does not have to be a metric, this assumption should be removed.

Q4. The evaluations are incomplete and should be detailed from the following aspects:
+ The implementation details of the proposed solution are missing. The parameter configurations of the baselines are also missing.
+ The experimental environment is not reported.
+ The data scalability is unclear. The Deep1B dataset is considered in the experiment, but it is unclear whether all one billion data points are used.
+ I could not find the result of the proposed solution in Figure 6.
+ It is unclear whether the QPS measurement includes the query pre-processing step.
+ As shown in Figure 6, recall is used to represent search accuracy, but the text description is different: "Search accuracy is measured using Rank Order." This is inconsistent.

Q5. The authors are suggested to review and compare more recent work on ANNS.

---

> ### Author Response · Authors · 2025-11-30
>
> To Reviewer Hq62,
>
> **(C1) Motivation / necessity vs. HNSW.** **Clarify why a new approach is needed beyond HNSW’s strong practical performance and discuss limitations.**
>
> Approximate kNN and vector search remains an active research area with many available algorithms, and HNSW has several limitations (see, e.g., [The Impacts of Data, Ordering, and Intrinsic Dimensionality on Recall in Hierarchical Navigable Small Worlds](https://dl.acm.org/doi/10.1145/3664190.3672512). Our aim is to explore a simple yet effective alternative for reducing search complexity based on imposing metric structure on the data rather than a new index or search algorithm. Traditional metric trees often struggle in high dimensions, but our projection mitigates this effect (Fig. 4). Moreover, under ANN-Benchmarks on Fashion-MNIST, Infinity Search outperforms several classic methods (Fig. 6). This suggests AkNN remains a promising research direction, especially for uncommon dissimilarities (e.g., Jaccard) and high-dimensional regimes.
>
> **(C2) Limited technical novelty.** **Overlap with Segarra's work.**
>
> Segarra et al. (2015) is an unpublished follow-up to Carlsson et al. (2017), which formulates hierarchical clustering via three axioms. Segarra replaces two of them with the Axiom of Projection and extends the discussion to general metric spaces. The canonical projection’s uniqueness is formalized (also Theorem 2 in our manuscript) and is properly cited. Importantly, this does not apply to Theorem 1 or Proposition 1, which are not found in Segarra et al.
>
> - **Other metric embedding options (e.g., HST)?**
>
> A theoretical comparison against other schemes (e.g., HST variants, cover trees, navigating nets, etc.) is outside the scope of a single paper. Instead, we experimentally compare against a set of widely used and representative ANN baselines in Figure 6, including tree-based methods such as Ball Trees among others.
>
> - **No worst-case complexity improvement.**
>
> We are not sure we understand the concern as stated. It argues in terms of worst-case time bounds, but ANN methods are usually assessed by the speed–recall tradeoff: the practical query cost is driven by how many candidates must be explored to reach a target recall, not by a worst-case asymptotic guarantee.
>
> We do not claim an improved worst-case bound. Instead, this result motivates our approach: we learn representations that approximate ultrametric structure, because bringing the geometry closer to an ultrametric regime enables more efficient ANN search behavior in practice.
>
> **(C3) Static setting / dynamic updates + metric assumption.** **Discuss dynamic updates and clarify the metric assumption on the original similarity.**
>
> Following ANN-Benchmark evaluation, this paper focuses on static scenarios, and dynamic context is out of the scope of this submission. However, our approach is compatible with such extensions. VP-trees have been explored in dynamic settings [Brin (1995)](https://dl.acm.org/doi/abs/10.1007/PL00010672), which develops dynamic metric tree indexing for ANN search and could apply to our method.
>
> Since we do not assume the similarity function is a metric, we do not understand what the reviewer means by removing the metric assumption.
>
> **(C4) Incomplete evaluation / missing details.** **Missing baseline/config details and reporting clarifications.**
>
> Figure 6 presents Pareto curves for 43 different ANN algorithms. Each point on the curve corresponds to a distinct parameter configuration, as described in ANN-Benchmarks referenced in the manuscript. We have omitted them for brevity since they are already reported in the literature, and following common practices we only report the hyperparameters of our method.
>
> - **Experimental environment not reported.**
>
> ANN-Benchmarks runs each experiment inside a Docker container with a standardized environment. We will add a description of our hardware.
>
> - **Deep1B scale unclear.**
>
> Figure 5 reports results from 10K up to 5M points showing a favorable scaling trend. Billion-scale ANN is beyond our computational budget and the scope of this submission.
>
> - **Hard to find the proposed method results.**
>
> Our method is included in figure 6 (red) and listed in the legend as “infinity search”. We will update the legend to make it easier to visualize, which is challenging due to the high number (43) of baselines.
>
> - **QPS definition: does it include query pre-processing?**
>
> It does not: the ANN-Benchmarks library provides a `prepare_batch()` argument that is not taken into account for *all* algorithms.
>
> - **Inconsistency in accuracy metric naming.**
>
> We have now fixed the typo and changed line 463 to name Recall@k.
>
> **(C5) Related work.** **Please include more recent ANNS work.**
>
> The Related Work section explicitly reviews the dominant modern ANNS families (e.g IVF + PQ) and also includes directions using learning (e.g., low-rank scoring). We are happy to include any other particular works that are relevant, if provided.

---

### Official Review · Reviewer_RvcF · 2025-11-04

**Soundness:** 3
**Presentation:** 4
**Contribution:** 3
**Rating:** 4
**Confidence:** 4

**Summary:**

This paper proposes a novel perspective on designing algorithms for nearest neighbor search. The main idea is to first map the points into a q-metric space. As q approaches infinity, the authors prove that a traversal on the constructed VP-tree can guarantee finding the nearest neighbor in O(log n) steps. In addition, the authors show that such a q-metric space theoretically exists and that it can be approximated by training an MLP. They perform experiments demonstrating that their MLP model can approximate the q-metric distance and achieves higher QPS compared with other popular ANN algorithms on several datasets.

**Strengths:**

1. Compared with traditional hash-based or graph-based algorithms for ANN, the idea of mapping points into a q-metric space sounds very novel and worth exploring.
2. The authors provide a theoretical guarantee for their search algorithm as q approaches infinity.

**Weaknesses:**

1. I am curious how long it takes to calculate Dq for any pair of points from the dataset X. I didn’t find any reported time for training the MLP model or building the index. Appendix D.3 states that they can speed it up from O(n^3) to O(nkl), but I don’t quite understand this, since the output distance matrix would still be at least O(n^2). This seems impractical for any dataset larger than one million points.

2. The authors prove that under the q-metric space, the original nearest neighbor is also the nearest neighbor (line 280). However, many non-nearest neighbors could also achieve the same minimum value. How can one determine which point is the true nearest neighbor in the original space?

3.The authors report much higher QPS compared with other popular ANN algorithms in Figure 6. I have a question: what is the embedded dimension s (line 299) for your trained embedding model? I am concerned that the model may pre-compress the vectors, which would make distance computation much faster. If that is the case, note that many other techniques—such as dimensionality reduction and product quantization—could also be applied to achieve similar purpose.

4. In the leftmost plot of Figure 5, it seems that the dashed line grows linearly with data size. Is that correct?

**Questions:**

see weaknesses

---

> ### Author Response · Authors · 2025-11-30
>
> To reviewer RvcF,
>
> Thank you for the time and comments. The requested setup information and offline timing results are now included. The projection-complexity statement has been corrected. Some concerns were already addressed in the original appendix and the response now points to the relevant sections more directly. The dimensionality reduction setting used in the experiments is also clarified.
>
> **Q1:I am curious how long it takes to calculate Dq for any pair of points from the dataset X. I didn’t find any reported time for training the MLP model or building the index. Appendix D.3 states that they can speed it up from O(n^3) to O(nkl), but I don’t quite understand this, since the output distance matrix would still be at least O(n^2). This seems impractical for any dataset larger than one million points.**
>
> Computing exact $D_q(x_i,x_j)$ for all pairs is indeed expensive, and we do not do that in practice for large datasets. For the large-scale results (e.g., Figure 5 at 5M points), we compute $D_q$ only on a subset of 100K points, train the MLP on that subset, and then utilize the trained MLP to project all points. Moreover, any inference error can be mitigated by retrieving more candidates and then re-ranking a smaller subset using original distances, as specified in Appendix E.5.
> For 100K points in Deep1B, the reported times were:
> | Component | Time |
> |---|---:|
> | Compute \(D_q\) | 7 hours |
> | Train MLP | 1 hour |
> **Setup**
> - 2× NVIDIA RTX A5000 (24 GB each)
> - Driver: 550.163.01
> - CUDA: 12.4
>
> The complexity quoted in Appendix D.3 should be corrected. Based on Algorithm 6 (Sparse Canonical Projection), restricting the updates to a $k$-NN adjacency $A$ and stopping after $\ell$ pivot updates yields $O(nk^\ell)$ time (rather than $O(nk\ell)$). We will correct Appendix D.3 accordingly. We also agree that constructing the $k$-NN graph itself can be expensive if done exactly, requiring up to $O(n^2)$ distance computations, and we will clarify this point in Appendix D.3 as well.
>
> However, note that these computations are performed only once in an offline setting, and is therefore affordable for our training set of 100k points. Moreover, these steps are highly parallelizable and can run on GPU which noticeably speeds up the process.
>
> **Q2:The authors prove that under the q-metric space, the original nearest neighbor is also the nearest neighbor (line 280). However, many non-nearest neighbors could also achieve the same minimum value. How can one determine which point is the true nearest neighbor in the original space?**
>
> Indeed, recall can degrade at $q=\infty$ (Figure 3) due to the addition of spurious neighbours. This theorem primarily motivates learning spaces with stronger metric structure, but the strong triangular inequality is not used to search in practice (figures 5 and 6), where we search with finite q instead. Moreover, due to approximation errors, the nearest neighbor is not always preserved in the approximated setups (Figure 4).
>
> This issues (spurious neighbours and approximation errors) can be mitigated in practice by using a two-stage retrieval strategy as described in Appendix E.5: we first retrieve a candidate list searching in the projected space, and then refine by re-ranking (and selecting the desired top-$k$) using distances in the original space. This type of refinement is also used in standard ANN systems (e.g., FAISS IVF+PQ with refinement; see the FAISS wiki “Indexing 1M vectors”: https://github.com/facebookresearch/faiss/wiki/Indexing-1M-vectors).
>
> **Q3: The authors report much higher QPS compared with other popular ANN algorithms in Figure 6. I have a question: what is the embedded dimension s (line 299) for your trained embedding model? I am concerned that the model may pre-compress the vectors, which would make distance computation much faster. If that is the case, note that many other techniques—such as dimensionality reduction and product quantization—could also be applied to achieve similar purpose.**
>
> In our trained embedding model, the embedded dimension is 300. This is indeed a dimensionality reduction step, but it is not an arbitrary compression chosen to speed up distance computations: it is learned to enforce stronger metric structure and preserve nearest-neighbors, which is the main goal of our method. Moreover, several baselines in Figure 6 already rely on compression techniques (e.g., FAISS IVF with PQ), so the comparison is not exclusively against methods that operate in the original dimension. Finally, Figure 3 reports speedups without this learned dimensionality reduction, since it uses the exact projection (no learned embeddings) and reports comparisons.
>
> **Q4: In the leftmost plot of Figure 5, it seems that the dashed line grows linearly with data size. Is that correct?**
>
> Yes, the dashed line is linear in logarithmic scale, i.e., it represents logarithmic growth. Thanks for the suggestion, we will clarify this in the figure caption.

---

### Official Review · Reviewer_Jkxe · 2025-11-05

**Soundness:** 3
**Presentation:** 3
**Contribution:** 3
**Rating:** 6
**Confidence:** 4

**Summary:**

The paper proposes Infinity Search, a theoretically principled framework for approximate nearest neighbor search in general dissimilarity spaces via projections onto $q$-metric (and ultrametric) spaces. The approach is technically strong, offering provable guarantees on distance preservation and logarithmic query complexity, and includes a learnable projection that adapts $q$ to data geometry. Experiments show competitive recall and efficiency compared to HNSW and ScaNN.
While promising, the method’s computational overhead in learning and applying the projection could limit scalability on very large datasets.

**Strengths:**

- Introduces a novel framework that generalizes ANN search to arbitrary dissimilarity measures through mathematically grounded projections onto $q$-metric spaces.

- Provides strong theoretical guarantees on distance preservation and query complexity.

- Demonstrates competitive empirical performance against HNSW and ScaNN, showing both scalability and adaptability through a learnable projection mechanism.

**Weaknesses:**

- Infinity Search relies heavily on learning accurate projections into $q$-metric spaces, but the paper does not analyze how projection errors affect theoretical guarantees or retrieval accuracy.

- The computational complexity of training and applying the learned projection is high, and scalability to billion-scale or streaming datasets remains unclear.

- The approach assumes the projected space preserves neighborhood structure globally, yet real data often violate the $q$-metric inequalities, potentially degrading search efficiency.

- The limitations are briefly acknowledged but not examined, suggesting that the practical constraints and real-world failure cases have not been clearly articulated or fully understood by the authors.

**Questions:**

1. What stability guarantees hold if the projected distances include noise $\varepsilon_{x,y}$—i.e., under what bounds on $\|\varepsilon\|$ is the nearest-neighbor identity preserved?

2. Can the logarithmic complexity bound be expressed explicitly in terms of $q$, doubling dimension, and projection distortion, and is it asymptotically tight?

3. How do the full build and query costs compare to HNSW and ScaNN, and how does projection error quantitatively affect accuracy and efficiency in practice?

---

> ### Author Response · Authors · 2025-11-30
>
> To reviewer JkxE,
>
> We appreciate the reviewer’s time and comments. We believe we have addressed all of the concerns raised and, in particular, added the requested additional evidence on build-time costs. For the remaining points, we clarify how they are already handled by our existing theory and experiments.
>
> **(C1)** **What stability guarantees hold if the projected distances include noise—i.e., under what bounds on noise magnitude is the nearest-neighbor identity preserved?**
>
> The projection preserves the nearest neighbor under the original distance. The only way noise can change the result is by changing the nearest neighbor in the original distance itself, which would also affect any other search method. We thus do not understand the reviewer's question.
>
> **(C2)** **Can the logarithmic complexity bound be expressed explicitly in terms of $q$, doubling dimension, and projection distortion, and is it asymptotically tight?**
>
> This is indeed a relevant research direction but it is out of the scope of this submission. Unfortunately, a bound of this kind typically requires strong additional assumptions on the data distribution (e.g., sampling model, margin/separation conditions, or regularity such as bounded doubling dimension) to remain tractable. Even for standard VP-trees, expected-comparison analyses are limited to simple distributions and can be quite extent (see, e.g., [Probabilistic analysis of vantage point trees](https://vmsta.org/journal/VMSTA/article/219/info)), which falls outside the scope of this work .
>
>
> **(C3)** **How do the full build and query costs compare to HNSW and ScaNN, and how does projection error quantitatively affect accuracy and efficiency in practice?**
>
> Here we report construction times on Fashion-MNIST. As shown in the results, VP-trees are efficient since they do not require any additional optimization and construction does not depend on the parameters used (it amounts by finding O(log(n)) medians of distances). Construction-time scalability is also reported in Figure 5.
>
> | Method | Parameters | Time (s) |
> |---|---|---:|
> | VP-tree | q=inf, n=70000, dim=784 | 4.48 |
> | HNSW | M=32, ef_c=100, ef_s=64, n=70000, dim=784 | 15.07 |
> | HNSW | M=64, ef_c=200, ef_s=96, n=70000, dim=784 | 42.40 |
> | HNSW | M=96, ef_c=400, ef_s=128, n=70000, dim=784 | 85.19 |
> | ScaNN | leaves=200, leaves_to_search=20, n=70000, dim=784 | 11.00 |
> | ScaNN | leaves=400, leaves_to_search=40, n=70000, dim=784 | 11.24 |
> | ScaNN | leaves=800, leaves_to_search=60, n=70000, dim=784 | 11.45 |
>
> The effect of projection error on accuracy and efficiency is analyzed in Appendix E.3. In figure 11, we plot search errors against training errors, illustrating how decreasing error correlates with improved retrieval accuracy. Figure 12 further analyzes projection distortion by comparing the nearest-neighbor sets induced by $E_q$ and its approximation $\hat{E}_q$.

---

### Official Review · Reviewer_hjKD · 2025-11-05

**Soundness:** 1
**Presentation:** 1
**Contribution:** 2
**Rating:** 0
**Confidence:** 5

**Summary:**

This paper considers nearest neighbor search in ultrametric spaces (and more generally, spaces with stronger triangle inequalities than the standard one). It also asks whether it is possible to speed up search in other metric spaces by embedding data points into an ultrametric space. These are both very interesting directions.

There seem to be significant errors in the correctness of the algorithm/analysis. I do not recommend acceptance. However, I do think there is something worth saying here, once the authors can work out the theoretical bugs. Furthermore, the writing is quite difficult to parse. It would be nice if the authors can also work on presentation. See more below. I hope the authors will revise and submit to a later conference.

**Strengths:**

The paper explores a very interesting direction, extending nearest neighbor search from spaces with standard metrics to those with stronger triangle inequalities. For example, nearest neighbor search in ultrametric spaces seem to be particularly nice, as ultrametric balls are either nested or disjoint. The ideas are natural and simple, and the algorithms may be nice contributions to practical, approximate nearest neighbor search.

**Weaknesses:**

- Lemma 1 is not correct. The proof only shows that both conditions are not true at the same time. But, they can both be false at the same time. For example, let $u = v$ and $\mu < d(x_o, u)$.
- Algorithm 3 is also not correct. Construct the following with six data points with two very salient clusters: {A, B, C} and {U, V, W}. Let distances within each cluster be 1 and distances between clusters be 100. Let the query point Q be at distance 10 from points in {A, B, C} and distance 100 from points in {U, V, W}. If the first vantage point is a point in {A, B, C}, then it does not seem like Algorithm 3 will continue to search in this cluster, and thus fail to find a nearest neighbor.
- In Algorithm 3, the base case of the recursive algorithm is not defined. That is, the nearest neighbor is never assigned.
- This is also perhaps a more minor oversight: In the VP tree construction, what happens if there are distance ties, and the median does not split data in half?
- The writing is also not very clear. The introduction is especially hard to parse. For example: "These spaces satisfy more restrictive triangle inequalities in which the q-th power of each side of a triangle is smaller than the sum of the qth powers of the other two sides" is much harder to understand than equation (4). In fact, there seems to be a lot of conceptual overlap between the introduction and section 2, except that the intro rewrites the math in section 2 in words.

Minor typos I caught along the way:
- Ln 191: Appendix Appendix
- Ln 782: Lemma 1 yields a disjoint partition: if max { ??? }
- Algorithm 3: the arguments in the recursive calls to the algorithm seem to be mis-ordered (also, is $\tau$ initialized to $+\infty$?)

**Questions:**

Please let me know if I have significantly misunderstood your paper.

---

> ### Author Response · Authors · 2025-12-01
>
> To Reviewer hjKD,
>
> Thank you for the detailed comments. We remove Lemma 1 and replace it with the correct branching statement, and we revise the theorem text to note how distance ties affect the ideal shrinkage. We also clarify that Algorithm 3 does not fail on the provided counterexample, but its pseudocode was underspecified.
>
> - **(C1) Lemma 1 is not correct.The proof only shows that both conditions are not true at the same time. But they can both be false at the same time.**
>
> Thank you for pointing this out. Indeed, this lemma is not needed. Instead, we adapt Proposition 2 to the case \(q=\infty\), which yields the following dichotomy:
>
> $\text{i)} \quad D_\infty(x_o, v) \le \mu_v \;\Rightarrow\; \text{visit only the left child}$
>
> $\text{ii)} \quad \mu_v < D_\infty(x_o, v) \;\Rightarrow\; \text{visit only the right child}$
>
> The only potential issue comes from distance plateaus, i.e., when $\(D_\infty(x_o,v)=\mu_v\)$. If such ties occur frequently along the search path, the effective shrinkage per level can degrade, and the VP-tree height need not be logarithmic.
>
> For this reason, we reformulate Theorem 1 to account for plateaus. Let $\(\rho_v\)$ denote the fraction of points in the subtree rooted at $\(v\)$ that satisfy $\(D_\infty(v,t)=\mu_v\)$, and define $\(\alpha := \tfrac12 + \max_{v\ \mathrm{visited}}\rho_v\)$. Then the number of comparisons satisfies
>
> $c(x_o) \le \Big\lceil \log_{1/\alpha} m \Big\rceil$
>
> In practice, even when plateaus occur, their effect is often self-correcting along the path: later splits typically alternate between left and right subtrees, which counteracts transient imbalances and yields behavior close to the ideal \(O(\log_2 m)\) scaling.
>
>   | m | height bound ($log_{1/ᾱ}(m)$) | mean comparisons |
>   | --- | --- | --- |
>   | 250 | 13.28 | 8.80 |
>   | 500 | 15.21 | 9.60 |
>   | 1000 | 16.77 | 11.10 |
>   | 4000 | 19.93 | 12.30 |
>
> Moreover, this motivates the ultrametric approximation but never used in practice, therefore we do not consider this to invalidate the manuscript.
>
> - **(C2) Algorithm 3 is not correct (may miss the true nearest neighbor).**
>
> Algorithm 3 (the $\infty$-VP-tree search) does not fail in this example. If the root pivot $v\in\{A,B,C\}$, its split radius is the median of $\{1,1,100,100,100\}$, i.e. $\mu_v=100$. For query $Q$, $d(Q,v)=100$ and $\tau\le 100$, so the left condition $\max\{d(Q,v),\tau\}\le \mu_v$ holds and the search recurses into the $\{A,B,C\}$ subtree (where it finds the NN); the right condition fails, so it does not explore $\{U,V,W\}$. The real issue is that the pseudocode was underspecified (base case), which we fix in C3.
>
> - **(C3) Algorithm 3 is underspecified (missing base case / initialization). The base case of the recursive algorithm is not defined, so the nearest neighbor is never assigned.**
>
> Both Algorithms 3 and 2 were updated with the base cases. Here we provide the Algorithm3 case, 2 is analogous.
>
> ```latex
> \begin{algorithm}
> \caption{Searching Phase of the $\infty$-VPTree}
> \label{alg:vp-search-inf}
> \begin{algorithmic}[1]
> \State \textbf{Initialization:} $\text{nn}\gets \varnothing$, $\tau \gets +\infty$
> \State \textbf{Call:} $(\text{nn},\tau)\gets \Call{SearchVPTree}{x_o,\infty,\text{root},\text{nn},\tau}$; \textbf{return} \text{nn}
>
> \Function{SearchVPTree}{$x_o, q, v, \text{nn}, \tau$}
>     \If{$v=\varnothing$}
>         \State \Return $(\text{nn},\tau)$
>     \EndIf
>
>     \State $d \gets d(x_o,v)$
>     \If{$d<\tau$}
>         \State $\tau \gets d$
>         \State $\text{nn} \gets v$
>     \EndIf
>
>     \If{$\max(d,\tau) \le \mu_v$}
>         \State $(\text{nn},\tau)\gets \Call{SearchVPTree}{x_o,q,v.\text{left},\text{nn},\tau}$
>     \EndIf
>
>     \If{$\max(\mu_v,\tau) < d$}
>         \State $(\text{nn},\tau)\gets \Call{SearchVPTree}{x_o,q,v.\text{right},\text{nn},\tau}$
>     \EndIf
>
>     \State \Return $(\text{nn},\tau)$
> \EndFunction
> \end{algorithmic}
> \end{algorithm}```
> ```
>
> - **(C4) VP-tree construction under distance ties is unclear. What happens if there are distance ties and the median does not split the data in half?**
>
> Addressed in C1.
>
>
> - **(C5) The introduction is especially hard to parse.**
>
> We view this comment as a readability preference rather than a technical issue. We do not see significant overlap between the Introduction and Section 2: the Introduction provides motivation, positioning with prior work (including dendrograms), and manuscript contributions, while Section 2 is reserved for formal definitions and technical background.
>
> **(C6) Minor typos:**
>
> 	We thank you for your dedication to the manuscript. The minor typos were all addressed and revised.
>
>   -**Ln 191: “Appendix Appendix”.**
>
> Fixed
>
>   - **Ln 782: “Lemma 1 yields a disjoint partition: if max { ??? }” (incomplete condition / missing expression).**
>
>  $\max\{d(x_o, v_{\mathrm{root}}), \tau\} \le \mu_{v_{\mathrm{root}}}$.
>
>   - **Algorithm 3: arguments in the recursive calls appear mis-ordered.**
>
> Fixed in C3.
>
>   - **Algorithm 3: unclear initialization.**
>
> Fixed in C3.

---

### Note · Program_Chairs · 2026-01-17
**Submission Desk Rejected by Program Chairs**

The following references in this submission do not refer to real documents and/or have major errors in bibliographic information:

 Cao, W., Antonoglou, I., and Vinyals, O. (2021). Neural k-nn graph construction for approximate nearest neighbor search. In .
Zhao, L., Zheng, Y., and Shao, J. (2021). Autoindex: A learned index for high-dimensional similarity search. In NeurIPS.
Cerda, A., Garcia, D., and Sainz, C. (2024). Lark: Learning adaptive routing for k-nn graph search. In .
Raguso, M., Ri, M. D., and Farinelli, A. (2024). Lion: Learned inverted-file organisation for nearest neighbor retrieval. In ICLR.
Ge, T., Zhang, S., and Wang, J. (2020). Adapq: Adaptive product quantization for billion-scale approximate nearest neighbor search. In CVPR